# 4D Single-particle tracking with asynchronous read-out single-photon avalanche diode array detector

Andrea Bucci [1,2], Giorgio Tortarolo[1,5], Marcus Oliver Held [1], Luca Bega[1], Eleonora Perego [1,6], Francesco Castagnetti [3], Irene Bozzoni[3,4], Eli Slenders [1] & Giuseppe Vicidomini [1] ✉

Single-particle tracking techniques enable investigation of the complex functions and interactions of individual particles in biological environments. Many such techniques exist, each demonstrating trade-offs between spatiotemporal resolution, spatial and temporal range, technical complexity, and information content. To mitigate these trade-offs, we enhanced a confocal laser scanning microscope with an asynchronous read-out single-photon avalanche diode array detector. This detector provides an image of the particle's emission, precisely reflecting its position within the excitation volume. This localization is utilized in a real-time feedback system to drive the microscope scanning mechanism and ensure the particle remains centered inside the excitation volume. As each pixel is an independent single-photon detector, single-particle tracking is combined with fluorescence lifetime measurement. Our system achieves 40 nm lateral and 60 nm axial localization precision with 100 photons and sub-millisecond temporal sampling for real-time tracking. Offline tracking can refine this precision to the microsecond scale. We validated the system's spatiotemporal resolution by tracking fluorescent beads with diffusion coefficients up to 10 $\mu m^2$/s. Additionally, we investigated the movement of lysosomes in living SK-N-BE cells and measured the fluorescence lifetime of the marker expressed on a membrane protein. We expect that this implementation will open other correlative imaging and tracking studies.

Fluorescence single-particle tracking (SPT) is a fundamental tool for investigating the functions of individual particles in biological environments[1]. As an example, it has been successfully applied to study virus infection mechanisms[2–4], surface protein trafficking[5–8], molecular motors dynamics[9,10], and anomalous diffusion and transport[11,12].

The diversity among the biological phenomena whose study requires SPT techniques has driven method development in multiple directions. Although many approaches have been proposed, no one has emerged as the gold standard. Offline image-based SPT is the first and most straightforward method: the sample is illuminated in a wide-field configuration, and the fluorescence emission is recorded with a megapixel matrix detector such as a complementary metal-oxide semiconductor (CMOS) or charge-coupled device (CCD) camera. Repeated exposures over time produce a set of images that are analyzed offline to extract the positions of the single particles, while the

[1]Molecular Microscopy and Spectroscopy, Istituto Italiano di Tecnologia, Genoa, Italy. [2]Dipartimento di Informatica, Bioingegneria, Robotica e Ingegneria dei Sistemi, University of Genoa, Genoa, Italy. [3]Non coding RNAs in Physiology and Pathology, Istituto Italiano di Tecnologia, Genoa, Italy. [4]Department of Biology and Biotechnology Charles Darwin, Sapienza University, Rome, Italy. [5]Present address: Laboratory of Experimental Biophysics, EPFL, Lausanne, Switzerland. [6]Present address: Centre for Integrative Genomics, Université de Lausanne, Lausanne, Switzerland. ✉e-mail: giuseppe.vicidomini@iit.it

trajectories are reconstructed by linking the localizations frame by frame[8,13–15]. This approach is capable of following many particles in parallel with a spatial resolution (or localization precision) between 20 nm and 40 nm given a budget of $10^3 - 10^4$ photons[8,16,17] but is severely limited to $\approx 10$ ms in time resolution by the imaging rate of the detector and to $2 \mu m$ in axial range due to the fixed illumination plane[17].

An alternative class of techniques is real-time single-particle tracking (RT-SPT). This approach aims to overcome the temporal resolution and spatial range limitations of offline SPT. Real-time SPT retrieves the position of only a single particle inside a small observation volume and follows its movement over time by shifting the observation volume via a closed feedback loop. Common RT-SPT techniques are implemented on customized laser scanning microscopes featuring one or more single-pixel detectors, such as single-photon avalanche diodes (SPADs) or photo-multiplier tubes (PMTs). Therefore, experimental raw data consists of one (or more) intensity time traces in which the particle position is encoded using structured detection or structured illumination. In the first approach, the emission is split among multiple detectors, whose spatial arrangement is engineered to allow the inverse calculation of the 3D particle's position from the intensity traces[18–20]. Structured illumination, on the contrary, investigates the particle's location by sequentially moving the focused excitation beam at a certain number of points along a specified trajectory around the particle. The ideas proposed across the years are diverse and include the use of a Gaussian beam swiping a circle, such as in orbital tracking[21–23], or illuminating points arranged in a tetrahedron[24], or moved along a knight's tour[25,26]. Implementations with multiple Gaussian beams have also been proposed to increase the spatiotemporal resolution[27]. More recently, in MINFLUX a doughnut-shaped beam is displaced in a triangular pattern, showing a more efficient localization precision for a given number of photons[28–31]. Indeed, given a certain photon budget, the localization uncertainty of any RT-SPT technique can be theoretically calculated depending on the combination of the illumination shape and the spatial arrangement of the sampling points[32]. Regardless of the actual implementations, all RT-SPT techniques are affected by one or more of the following issues: high photon fluxes requirement, poor axial range, and technical complexity. Furthermore, the information about the photon emission time is often not exploited, despite being available in many cases. This prevents, for example, access to the fluorescence lifetime measurement, which is a powerful tool for investigating the particle's interactions and chemical nano-environment[33,34]. A more detailed discussion about the state of the art of RT-SPT can be found in the comprehensive review by van Heerden et al.[35].

In recent years, companies, as well as research groups, have directed growing attention toward the realization of better detectors. The development of new cameras (such as Hamamatsu qCMOS, Photonscore LINcam, and event-based sensors[36,37]) aims to improve the temporal resolution and information content of wide-field microscopy, including offline SPT. In the same way, advances in SPAD technology, and in particular the development of asynchronous read-out SPAD arrays, are beneficial for laser scanning microscopy techniques[38,39]. An asynchronous read-out SPAD array detector is constituted by a set of independent SPADs placed on the same chip with a predetermined spatial arrangement. Each pixel works as a standalone detector with single-photon sensing and time-tagging capabilities. It is directly wired to its output channel, which fires a digital signal with high (a few hundred picoseconds) temporal precision for each detected photon. In such a fashion, the detector can be combined with a multichannel time-resolved data-acquisition system (e.g., a photon time-tagging module) to provide spatial information like a small camera while overcoming the frame rate limitation. Indeed, the camera's frame rate is theoretically infinite by having a completely pixel-based asynchronous read-out. Only the so-called pixel dead time

practically limits the frame rate: after detecting a photon, the pixel is blind for a few tens of nanoseconds. In addition, every photon can be tagged with its emission time with respect to the excitation laser pulse, thus giving access to the fluorescence lifetime information. SPAD array detectors have already been employed to increase the resolution and contrast of confocal images through image scanning microscopy[40–45] and to enhance the information content and flexibility of fluorescence correlation spectroscopy[46,47]. Both imaging and spectroscopy approaches have been synergistically combined with fluorescence lifetime analysis[48].

Here, we present a novel feedback-based RT-SPT implementation based on a common laser scanning microscope equipped with a $5 \times 5$ asynchronous read-out SPAD array detector and an astigmatic detection. The detection scheme enables direct and almost instantaneous 3D localization of particles within a relatively small volume, in the order of the size of the sub-micrometer focal excitation volume, in all directions ($x, y, z$). This information enables dynamic repositioning of the beam scanning system to keep the particle centered in the excitation volume. As a result, the effective tracking range of real-time single-particle tracking is primarily constrained by the lateral and axial scanning capabilities of the microscope. Furthermore, the SPAD array detector simultaneously measures the particle's fluorescence lifetime $\tau$.

As such, our RT-SPT implementation effectively produces a 4D trajectory in time and space $[x(t), y(t), z(t), \tau(t)]$, thus improving the information content while reducing the architectural complexity in comparison to any approach mentioned above. We call this implementation real-time 4D single-particle tracking (RT-4D-SPT). Here, there is no need to perform orbital scanning or other fast excitation beam shifting to encode the single particle's position. Notably, the localization of the particle – for the re-centering of the excitation volume – is calculated in real time by the data-acquisition (DAQ) system without introducing a sensitive delay in the feedback closed loop system. The spatiotemporal resolution of the proposed RT-4D-SPT approach is practically limited only by the flux of the detected photons and by the lag-time of the laser beam scanning system, for re-centering onto the particle before it escapes from the excitation volume. In addition, the full SPAD array signal is transferred to the computer to refine the particle trajectory with offline algorithms.

We validate our technique by comparing the localization uncertainty obtained from the Cramér-Rao bound calculations with the experimental localizations of fixed fluorescent particles. We then track fluorescent particles in two configurations: moved along a pre-determined path and freely diffusing in water or a viscous glycerol solution. To prove the versatility of our technique, we apply it in a biological context to investigate the movement of lysosomes in living cells along microtubule filaments while assessing the lifetime of the green fluorescent protein (GFP) marker expressed on a protein on their membrane.

## Results

### Principle of real-time 4D single-particle tracking

To perform RT-4D-SPT, we use a 3D laser scanning microscope with minimal modifications (Fig. 1a). The fluorescence emission generated from the diffraction-limited excitation volume at the scanning position $\mathbf{r}_s = (x_s, y_s, z_s)$ is made astigmatic by a cylindrical lens and focused onto the SPAD array detector located into a conjugate plane, which replaces the traditional confocal pinhole and the single-pixel detector. In this context, it's crucial to emphasize that the SPAD array detector records the fluorescent signal in de-scanned mode, ensuring that the excitation volume and the static field-of-view (sFoV) (i.e., the region of the sample filling the sensor area) are consistently co-aligned. Our SPAD array detector is a specialized device that combines the temporal performance of a SPAD with the structured detection of a $5 \times 5$ camera, imaging a small sFoV (695 nm about $\approx 1.4$ A U). When a photon is

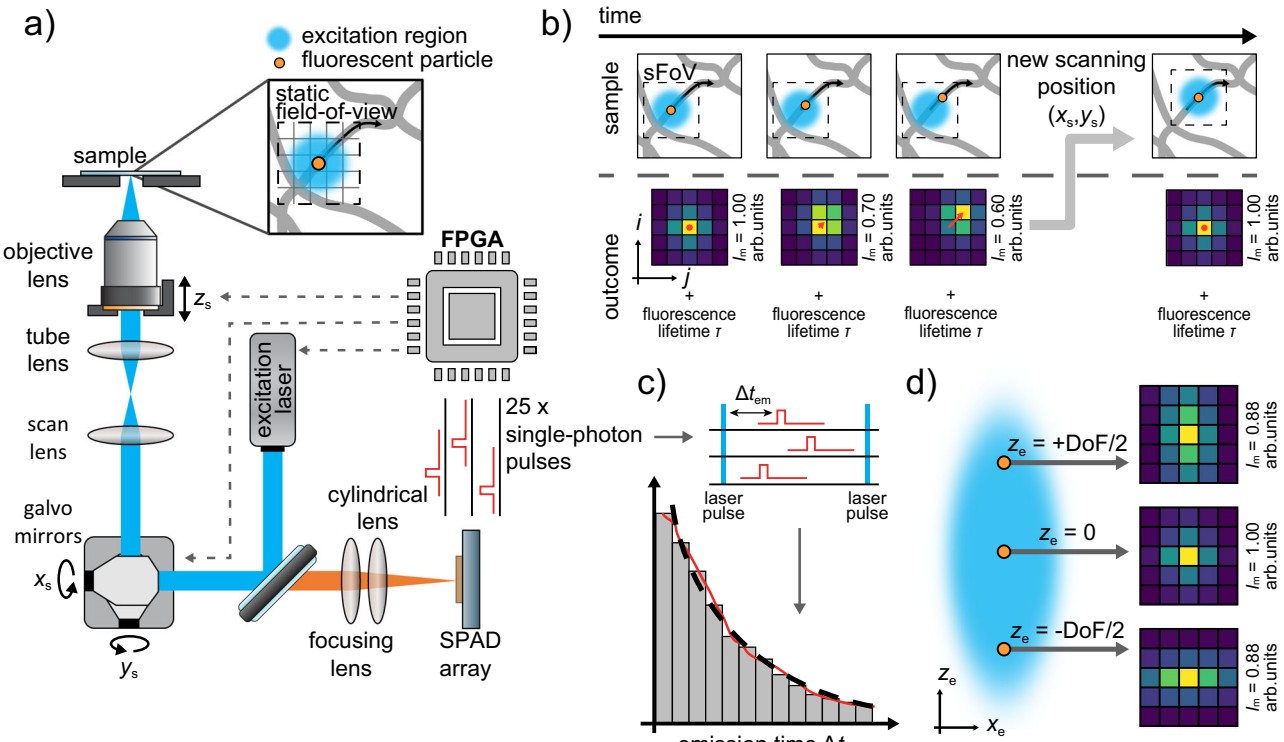

**Fig. 1 | Real-time 4D single-particle tracking on a laser scanning microscope equipped with a SPAD array detector. a** The optical setup is based on a 3D laser scanning microscope, where the confocalized detection unit is substituted with a SPAD array detector, and a cylindrical lens is inserted. The detector sFoV ($\approx$ 1.4 A U) is composed of 5 × 5 elements leading to 25 independent single-photon pulse trains. The FPGA receives the signals, calculates the emitter position $\mathbf{r}_e$, and updates the 3D scanning position $\mathbf{r}_s$ in real time. **b** A 2D movement of the single particle with respect to the center of the sFoV results in a shift of the detected emission pattern and a reduction in its intensity $I_m$, as shown in the simulated microimages. The detector also allows the concurrent measurement of the fluorescence lifetime $\tau$.

When the re-centering condition is triggered, a new position $(x_s, y_s)$ is estimated, and the sFoV is re-centered onto the particle with the beam positioners. **c** The temporal performance of each SPAD element of the array detector is similar to the single element counterpart (time jitter < 200 ps). Thus, the fluorescence lifetime of the emitter can be extracted by analyzing the emission time histogram $\Delta t_{em}$ obtained as the delay between each pulsed excitation at time $t_{exc}$ and the associated photon detection events at time $t_d$. **d** The quantitative estimation of the axial position of the particle leverages the astigmatism caused by the cylindrical lens. In fact, the symmetry and shape of the emission pattern registered in the microimage changes throughout the depth-of-field (DoF).

detected by a sensitive element of the array at coordinates $(i, j)$, it produces a short electric pulse in the corresponding output channel, with a time jitter below 200 ps. The signal is fed into a multi-channel time-resolved DAQ system, which allows for measuring the photon detection time $t_d$. Thus, each detection event corresponds to an independent 6-dimensional data point $(x_s, y_s, z_s, i, j, t_d)$.

This dataset is used to estimate the emitter position within the sFoV $\mathbf{r}_e = (x_e, y_e, z_e)$. To obtain $\mathbf{r}_e$, we can utilize position estimators designed for wide-field microscopy, such as the centroid estimator. This estimator entails low computational costs, hence can be easily implemented on the DAQ board, leveraging the field-programmable gate array (FPGA). As a result, a rapid feedback loop is employed for re-centering the excitation volume, and, consequently, the sFoV (Fig. 1b). The centroid estimator analyzes the 5 × 5 images $I(i, j)$, referred to as microimages, obtained by accumulating the detected photon stream within a specific and tunable temporal window. Any particle movement produces a distinguishable change in the intensity distribution in the microimage. Specifically, a lateral displacement leads to a shift in the distribution's center while, by making the detection astigmatic, its shape is uniquely linked to the axial position[49–51] (Fig. 1d). The same FPGA-based DAQ board acts as a control unit for the whole microscope and its beam scanning apparatus: a pair of galvanometer mirrors for the lateral displacement and a piezo objective for the axial shift. The control unit delivers the input signals to the beam scanning devices to displace the sFoV in the new position.

In short, our RT-SPT method leverages the spatial information provided by the SPAD array to continuously re-center the sFoV onto the particle. Like any RT-SPT method, our approach can effectively track a particle only if the localization of its position is precise and the re-centering of the sFoV is fast enough to prevent the particle from escaping the sensitive region. Indeed, the localization uncertainty scales with the number of photons detected, whose emission, in turn, requires a certain period of time depending on the particle's brightness. Because the SPAD array detector has practically sub-microsecond temporal resolution and, for a given microimage, the FPGA can calculate the particle's position in less than 100 ns, the spatiotemporal resolution — i.e., distinguishing two positions of the same particle both in time and space[52] — depends mostly on the lag time of the actuators responsible for re-centering the sFoV and the brightness of the particle.

Importantly, the SPAD array detector transfers microimages to the PC at a rate significantly higher than the real-time re-centering rate. If brightness is not the limiting factor, the spatiotemporal resolution can, therefore, be enhanced with offline analysis. We can apply more robust and precise estimators, such as maximum likelihood, to enhance the localization uncertainty (spatial resolution) or integrate the microimages to a higher localization rate (temporal resolution). However, it is crucial to consider that a higher localization rate comes at the expense of lower photon counts, generally resulting in lower localization precision.

Regarding the spatial range of our RT-4D-SPT method, while the sFoV is confined to a few hundred nanometers, the effective tracking range can vary significantly, spanning orders of magnitude depending on the effective scanning capabilities of the microscope. In the case of high numerical objective lenses such as our implementation, these ranges can extend over a few hundred micrometers laterally and around a hundred micrometers axially. However, it's crucial to acknowledge that optical aberrations, encompassing field curvature and spherical aberration along the lateral and axial directions, respectively, may substantially reduce these practical tracking values.

Not less important than deciphering the particle's position is studying the particle's fluorescence lifetime as a function of time. By using a pulsed excitation laser and implementing a series of fast time-to-digital converters (TDCs) directly in the DAQ system, we can obtain the delay between the excitation event and the photon detection, the so-called photon emission time, $\Delta t_{em} = t_d - t_{exc}$. This information enables us to calculate the fluorescence lifetime of the particle $\tau$ – potentially in real-time (Fig. 1c). In this work, we implemented within the FPGA-based DAQ system 25 TDCs by using a digital frequency domain (DFD) architecture. With respect to a more precise implementation based on tapped-delay lines, the DFD architecture requires less FPGA resources and sustains the maximum photon flux achieved by the SPAD array detector. Specifically, our DFD implementation provides a sampling step of 397 ps, a value lower than the tens of picoseconds obtained by using TDC based on tapped-delay lines, but still optimal for fluorescence lifetime applications. Furthermore, the DFD implementation is optimal when working at high photon flux rates[53]. The low requirements in terms of FPGA resources allow for synergistic integration of the multiple TDCs in the same DAQ module, which controls the tracking microscope, providing a compact and comprehensive architecture.

## Localization within the static field-of-view

As described above, in RT-SPT, the ability to track a single moving particle depends on the precise, accurate, and timely localization of its position. This enables the re-centering of the excitation volume – and thus the sFoV – on the particle before the particle leaves the sensitive region. Consequently, the characterization of the localization performance stands as a critical first step to thoroughly assess our RT-SPT performance. In particular, it is crucial to define the region in which – for a given signal-to-background ratio – the particle can be reliably localized, which we named the optimal localization volume (OLV). Evidently, the OLV does not necessarily coincide with the sFoV as it is also influenced by the dimension and shape of the excitation volume: the photon flux from the particle decreases moving away from the center of the Gaussian excitation volume, leading to a decrease in localization precision. Here, we quantitatively determine the OLV by measuring the localization precision for our RT-4D-SPT approach.

According to estimation theory, one way to quantify the theoretical lower limit of the localization uncertainty is to calculate the so-called Cramér-Rao bound (CRB)[54]. The CRB is a statistical measure that provides a theoretical lower bound on the variance of any unbiased estimator of an unknown parameter. In our case, the unknown parameter is the 3D position of the particle being tracked. To calculate the CRB, we adapt the mathematical framework developed by Balzarotti et al. for MINFLUX[28] and Masullo et al. for sequential structured illumination single-molecule localization microscopy[32]. Specifically, we introduce some minor modifications to account for our structured detection scheme (Supplementary Information Note 1). Our calculation relies on three crucial assumptions: (1) both the signal and background photon counts follow a Poisson distribution, (2) the signal exhibits linear dependence on the excitation light intensity, and (3) the background is independent of the particle's position. The first assumption is applicable to SPAD detectors, the second necessitates low excitation to avoid fluorescence saturation, and the third

condition is typically met when the background includes a combination of detector dark counts and unwanted signals from the sample, such as scattering, autofluorescence, and out-of-focus fluorescence. Considering a fixed acquisition time, the total number of photons detected by the sensor $N(\mathbf{r}_e)$, and consequently the signal-to-noise ratio $SBR(\mathbf{r}_e)$, are dependent on the position $\mathbf{r}_e$ of the emitter relative to the center of the excitation volume – equivalently the sFoV. However, by applying (2) and having measured the point spread functions (PSFs) for each detector element, we can fully characterize any experimental condition simply with the scalar parameters $N_p = N(\mathbf{0})$ and $SBR_p = SBR(\mathbf{0})$. Henceforth, these parameters will be extensively employed as benchmarks.

We perform the CRB calculations using the experimental PSFs measured with 20 nm fluorescent beads (Supplementary Fig. 1). Assuming $N_p = 100$ photons and $SBR_p = 5$, we obtain the maps of the CRB in the lateral ($\sigma_{xy}^2 = \sigma_x^2 + \sigma_y^2$) and axial ($\sigma_{xz}^2 = \sigma_x^2 + \sigma_z^2$) planes (Fig. 2a, d). From the isoline curves, we can identify a volume of approximately 300 nm × 300 nm × 500 nm in which we expect an approximately flat localization uncertainty with a maximum value between 40 nm and 60 nm. Amongst all the localization estimators proposed by different techniques through the years, the maximum likelihood estimator (MLE) has emerged for its reliability and solid mathematical background[49,55,56]. Remarkably, it provides an unbiased and linear estimation (Supplementary Information Note 2), and it is demonstrated to be fully efficient, which means it asymptotically reaches the CRB.

The high computational complexity necessary to perform the estimation with the MLE makes it the ideal choice for offline analysis but hinders its implementation in the real-time loop. We consequently need a new set of faster estimators, which allows us to gain computational speed at the expense of precision. We, therefore, introduce the centroid and normalized difference estimators for the lateral and axial localization, respectively. Compared to the MLE estimator, both the centroid and the normalized difference are computationally faster to evaluate but less resistant to noise, and their linearity is affected by the SBR. Furthermore, the normalized difference estimator for the axial localization is not independent of the lateral position of the emitter (Supplementary Information Note 2).

To compare the planar localization uncertainty of each estimator with the theoretical lower limit, we localize a single 20 nm fluorescent bead replicating the emitting conditions of the CRB calculations. Using the 3D piezo stage, the particle is shifted across the entire sFoV and localized offline multiple times with both the MLE and the faster estimators at each position. We then take the standard deviation as a measure of the localization uncertainty to plot the lateral (Fig. 2b, c) and axial (Fig. 2e, f) planar uncertainties for the MLE (Fig. 2b, e), the centroid (Fig. 2c), and the normalized difference (Fig. 2f) estimators.

The comparison of the MLE results with the theoretical lower limit shows a slightly worse performance in the global minimum value, especially in the lateral plane. This effect can be justified by considering the additional noise sources that may occur during the experiment but are not included in the CRB model, such as sample and microscope drifts and vibrations (Supplementary Fig. 12) and other sources of background which further degrade the SBR. As expected, the approximated nature of the faster estimators causes a further deterioration of their performance, in our cause by approximately 15%. By looking at the line profiles (Supplementary Fig. 13), we observe the overall shape of both the lateral and axial uncertainty maps are nevertheless consistent with the CRB. We define the OLV as the region in which the uncertainty increases at most 50% above the global minimum. Hence, we identify a volume of at least 300 nm × 300 nm × 600 nm inside which the performance of the localization is considered optimal with all the estimators. In particular, the faster estimators provide a lateral planar uncertainty between 32 nm and 53 nm and an axial planar uncertainty between 50 nm and 80 nm.

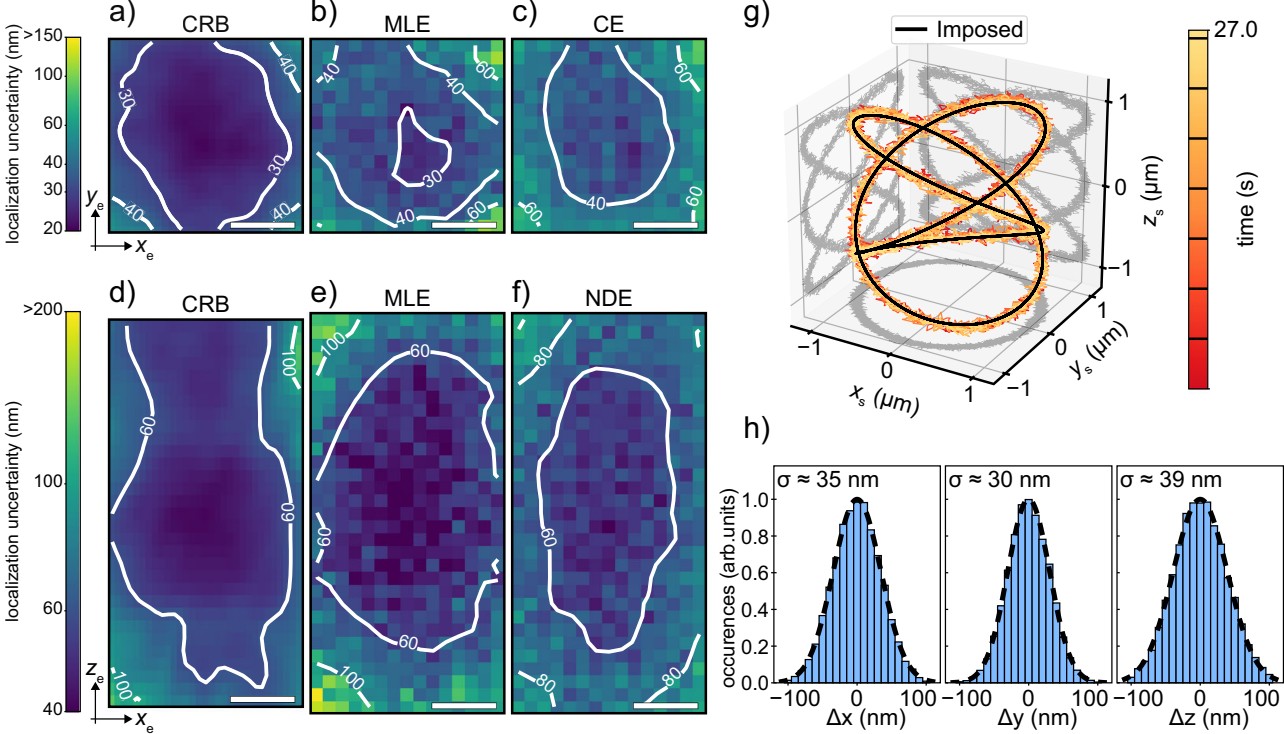

**Fig. 2 | Characterization of the planar localization and tracking uncertainties.**
**a–c** Lateral localization uncertainty maps $\sigma_{xy}(x_e, y_e)$. The CRB is calculated using an experimental PSF, $SBR_p = 5$ and $N_p = 100$ photons, and assuming a fixed acquisition time. The uncertainty maps of the MLE and centroid estimator are measured by using a common dataset obtained by scanning 20 times a 20 nm fluorescent bead replicating the same conditions of the CRB. $\lambda_{exc} = 561$ nm. Scale bar = 100 nm.
**d–f** Axial localization uncertainty maps $\sigma_{xz}(x_e, z_e)$. The CRB is calculated using an experimental PSF, $SBR_p = 5$ and $N_p = 100$ photons, and assuming a fixed acquisition time. The uncertainty maps of the MLE and normalized difference estimator are measured by using a common dataset obtained by scanning 33 times a 20 nm fluorescent bead replicating the same conditions of the CRB. $\lambda_{exc} = 561$ nm. Scale

bar = 100 nm. **g** Imposed and tracked trajectories of a single 20 nm fluorescent bead ($\lambda_{exc} = 561$ nm) moved at an average tangential speed of 5.5 µm/s along a 3D Lissajous pattern. The scanning position is updated simultaneously in the lateral direction with the centroid estimator and in the axial direction with the normalized difference estimator every 100 photons ($\langle\Delta t_{rc}\rangle = 1.9 \pm 0.4$ ms). The color is used to visualize time, and black lines on the colormap mark the end of each complete turn of the periodic pattern. Extended time trace of the trajectory in Supplementary Fig. 14. **h** Histograms of the difference between the imposed and tracked trajectories along the three directions for the experiment in (**g**). $\sigma_x = 34.60 \pm 0.04$ nm, $\sigma_y = 30.36 \pm 0.04$ nm, and $\sigma_z = 39.13 \pm 0.07$ nm.

Considering the rise time of the positioners ($\approx 200$ µs laterally and $\approx 4$ ms axially) together with the dimension of the OLV, we predict a maximum measurable diffusion coefficient for Brownian motion of $\approx 10$ µm²/s (Supplementary Information Note 3) – assuming that particle brightness is not a limitation. This upper limit matches with the diffusion coefficients of relatively small proteins (above $30 - 40$ kDa) moving in the cytoplasm and the membrane of cells[57–59], while it may be unsuitable for diffusion of the same protein in pure water[60].

### Real-time 3D and 4D tracking

The tracking procedure leverages the localization information to keep the single particle always in focus by updating in real-time the beam scanning position ($x_s, y_s, z_s$). To test its performance in a controlled setting, we first track a fixed 20 nm fluorescent bead moved along an imposed 3D Lissajous pattern at an average speed of 5.5 µm/s (Fig. 2g and Supplementary Fig. 14). As already anticipated, due to timing constraints, we rely on the centroid and normalized difference estimators to retrieve the particle position $\mathbf{r}_e$. These are less precise than the MLE and potentially non-linear. To assess whether these drawbacks affect the performance of the feedback loop, we calculate the difference between the imposed and tracked trajectories (Fig. 2h). By fitting the distributions with Gaussian functions, we obtain the lateral and axial planar tracking uncertainties relative to the detection of 100 photons: $\sigma_{xy} = 46.03 \pm 0.04$ nm and $\sigma_{xz} = 52.23 \pm 0.06$ nm. These values are comparable to the planar localization uncertainties of the fast estimators and consistent with localizing a particle within the central

region of the OLV. Remarkably, despite potential noise from positioner jitter and inertia, the tracking uncertainty does not significantly worsen. This suggests that the estimation process remains the dominant source of uncertainty in the feedback loop. In addition, the experiment confirms that the loop is correctly implemented and effectively keeps the particle within the OLV.

Because of the single-photon asynchronous read-out of the SPAD array detector, the re-centering of the sFoV can be decided a-priori at a specific frequency, i.e., every fixed time interval $\Delta t_{rc}$ – as for synchronous detectors with limited frame rate – or, more interestingly, when a specific event occurs. For example, when the number of detected photons reaches a particular value, namely when a suitable SBR is obtained – as for the previous RT-SPT experiment. This capability enables us to easily set the tracking conditions to match the brightness of the particle, thus achieving the best real-time temporal resolution. A further interesting feature of our RT-SPT implementation is the ability to decouple the lateral and axial re-centering by imposing different update rates or different target photon countings for the two dimensions. This allows achieving an isotropic localization precision at the expense of anisotropic re-centering times.

We can now eliminate the constraint of a ground truth, such as an imposed trajectory, to characterize our technique in a more realistic scenario. To achieve this, we opt to track fluorescent beads of three different sizes (40, 100, and 200 nm in diameter) freely diffusing in water. These sizes are specifically chosen to validate our previous estimation of the maximum measurable diffusion coefficient. Despite

being a single-particle technique, hence primarily focused on generating individual particle trajectories, our method can be automated and optimized to sequentially capture multiple trajectories from the same sample. This is accomplished by programming the control module to initiate tracking whenever the photon flux surpasses a user-defined threshold and to continue tracking as long as this condition is met. Subsequently, the system remains stationary until a new particle is recognized, and the process is repeated iteratively (further details in "Methods"). By tuning the tracking threshold, excitation intensity, and particle concentration, we achieve a condition in which we find, track, and lose sight of a new bead within seconds. This enables us to collect a dataset of hundreds of tracked beads for each size in tens of minutes (Supplementary Fig. 15). Each trajectory is analyzed independently to generate a distribution of diffusion coefficients (Fig. 3a). As expected,

smaller particles yield larger diffusion coefficients and, collectively, the histograms cover a range of coefficients up to and exceeding 10 μm²/s. The reliability of the trajectories obtained with our SPT technique is further validated by deliberately changing the average diffusion coefficient. For a fixed bead size, this is achieved by controlling the viscosity through the addition of glycerol at varying concentration levels (Supplementary Information Note 4). Furthermore, our system's capabilities extend beyond 3D spatial tracking and enable simultaneous measurement of each particle's fluorescence lifetime, $\tau(t)$. The FPGA control module manages the synchronization of pulsed excitation and photon detection, generating real-time histograms of the photon emission time. To achieve this, we implement 25 parallel TDCs, one for each SPAD pixel, employing the DFD principle[61,62]. Specifically, our implementation produces 25 fluorescence lifetime – or photon

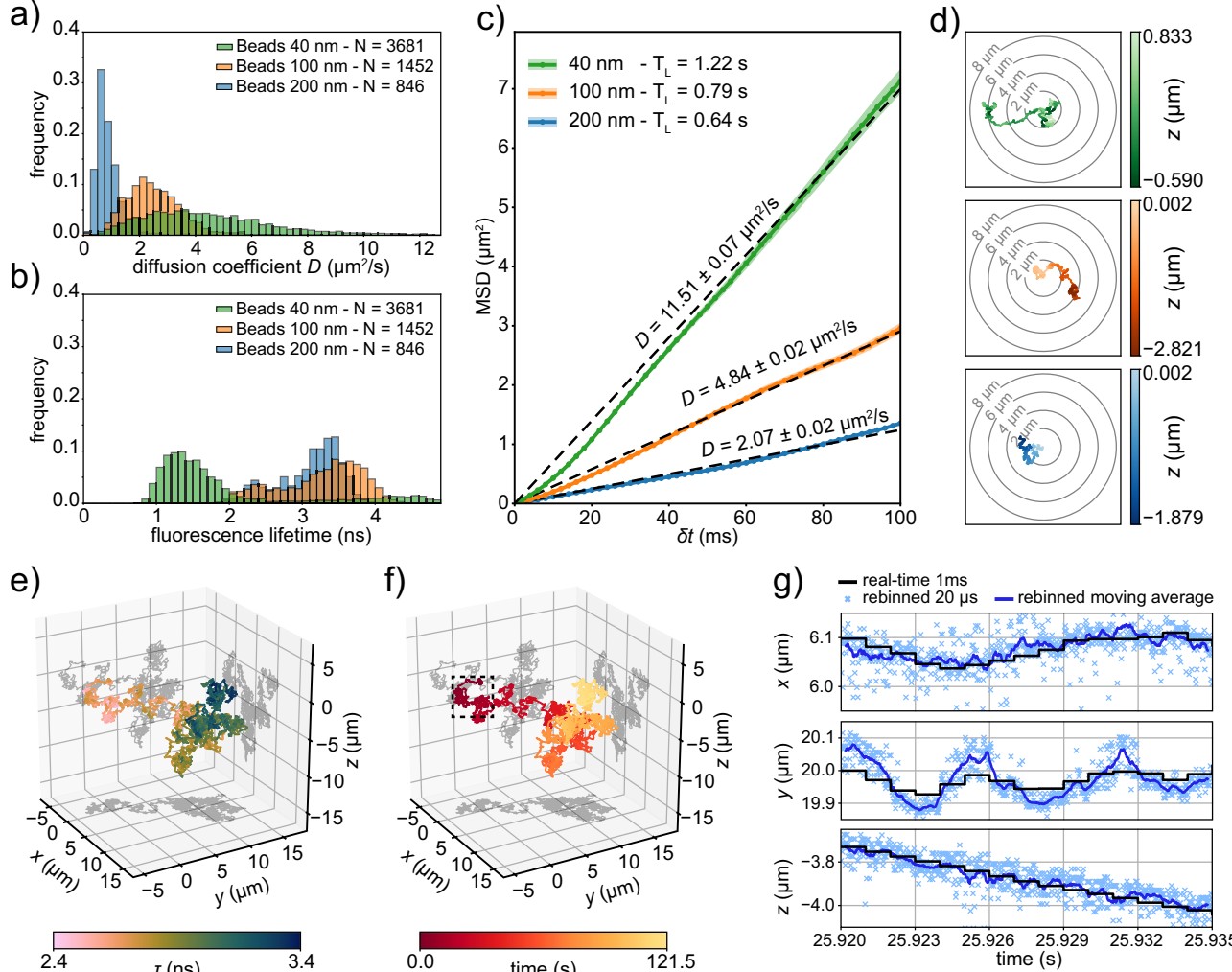

**Fig. 3 | 4D tracking on free fluorescent beads in water. a** Distributions of the measured diffusion coefficients for different fluorescent beads ($\lambda_{exc} = 488$ nm) freely diffusing in pure water. The legend reports the number of single trajectories acquired per bead size. For each single bead trajectory, the re-centering in the lateral and axial directions is performed at a fixed timing of $\Delta t_{rc}^{lat} = 1$ ms and $\Delta t_{rc}^{ax} = 2$ ms respectively. Cumulative distribution function in Supplementary Fig. 15a. **b** Distributions of the fluorescence lifetimes for the same fluorescent beads populations in (**a**). Each entry is calculated as the average fluorescence lifetime of a single bead trajectory. Differences in fluorescence lifetime between the populations may be due to variations in the fluorophore concentration inside the beads. Cumulative distribution function in Supplementary Fig. 15b. **c** Example of the MSD of three single trajectories from the beads populations in (**a**). The shaded area is the standard error of the mean. Each curve is fitted with a linear model, and the relative

diffusion coefficient $D$ is displayed accordingly. The legend reports the tracking time length $T_L$ of each trajectory. **d** 2D projections of the spatial trajectories for the beads in (**c**). The color visualizes the axial direction. For a fair comparison, each trajectory is cropped at a duration of 0.5 s. Extended time trace of the trajectories in Supplementary Fig. 16. **e** 3D representation of the trajectory of a single 200 nm fluorescent bead ($\lambda_{exc} = 488$ nm) diffusing in pure water. The color visualizes the fluorescence lifetime $\tau$. The re-centering in the lateral and axial directions is performed simultaneously at $\Delta t_{rc}^{all} = 1$ ms. The estimation of the lifetime is performed every $\Delta t_{\tau} = 10$ ms. Extended time trace of the trajectory in Supplementary Fig. 17. **f** 3D representation of the same trajectory as (**e**), but with color used to visualize the elapsed time. **g** Detail of the spatial trajectory of (**e**) and (**f**) rebinned in all the directions in postprocessing with a dwell time of 20 μs. The localization is also refined by using the MLE.

emission time – histograms every ≈ 5.7 µs with a sampling step of 397 ps[53]. Consequently, alongside the diffusion coefficient, we also obtain distributions of the average fluorescence lifetime for the same bead populations (Fig. 3b). From the entire dataset, we select three exemplar 4D trajectories, one per bead size, characterized by a tracking time length $T_L$ exceeding 0.5 s and a relatively low photon flux. The linear nature of the mean squared displacement (MSD) plots indicates Brownian motion (Fig. 3c), with the diffusion coefficients consistent with the estimated values (see Supplementary Eq. S.20). Coherently, for a fixed time interval, a higher diffusion coefficient correlates with a longer diffused range (Fig. 3d and Supplementary Fig. 16).

When the scientific question is directed towards the behavior of a single specific particle rather than acquiring a statistical population, as often seen in cell trafficking studies, adjusting the tracking settings becomes crucial to prioritize longer trajectory measurements. This involves reducing the tracking threshold and, when feasible, enhancing the SBR. With respect to the previous acquisitions, we extend the tracking duration of a single 200 nm fluorescent bead freely diffusing in water to over 2 min by leveraging its remarkable photostability even at high photon fluxes. A first intuitive representation of the 4-dimensional trajectory $Trj(t) = [x(t), y(t), z(t), τ(t)]$ focuses on the spatial information by plotting the 3D trajectory $[x(t), y(t), z(t)]$ in space and adding another observable through color coding (Fig. 3e, f and Supplementary Fig. 17). We notice the fluorescence lifetime seems to exhibit a space-dependent behavior, with a region near the beginning of the trajectory associated to a value of around 2.5 ns which then fades to 3.4 ns as the particle diffuses. However, in this specific example, a deeper understanding is obtained by observing the time evolution of each component of the 4D trajectory (Supplementary Fig. 17). In particular, the fluorescence lifetime track reveals the presence of self-quenching, which causes a decrease in a lifetime when a high amount of fluorophores are packed in proximity to each other, i.e., in a big fluorescent bead. As some emitters get bleached, this condition relaxes, and the fluorescence lifetime gradually reaches the value associated with the fluorophore in isolated conditions. Simultaneously, the fluorescence efficiency of the remaining fluorophores increases, which justifies the lack of a concurrent intensity drop. Furthermore, the diffusion regime is studied by calculating the MSD curve and the power spectral density (Supplementary Fig. 18), from which we deduce the particle is undergoing Brownian motion with a diffusion coefficient of ≈ 0.5 µm²/s.

Notably, this single experiment demonstrates the benefits of having a 4D dataset, which allows us to simultaneously reveal spatial properties, such as the diffusion regime, and photophysical properties encoded in the lifetime, such as self-quenching.

## Offline tracking and postprocessing

Our RT-4D-SPT technique belongs to the macro area of real-time approaches, but the information content of the data is richer than its usage in real-time and eventually emerges with postprocessing analysis, as typically happens in offline tracking. To understand the spatio-temporal resolution potentially achievable by our system, we consider its hybrid real-time/offline nature.

In the context of real-time processing, although the read-out of the SPAD array pulses occurs every 2 ns, the computational steps required to produce a re-centering response demand significant computational resources (see the pipeline described in "Methods"). We experimentally measured that a triggering input on the digital card is translated into a voltage output on the analog card with a minimum time delay of 400.0 ± 2.5 ns. However, the actuation is further delayed by the rise time of the positioners (Supplementary Information Note 3), necessitating approximately ≈ 200 µs for the galvanometric mirrors (150 nm lateral shift) and ≈ 5 ms for the objective piezoelectric stage (300 nm axial shift). Consequently, each real-time trajectory can be assumed to have a temporal resolution in the millisecond regime.

While the rise time of the positioners imposes a limitation on maximum temporal performance in real-time, the effective time resolution of our technique remains independent of the maximum trackable speed, as the control module sends the data to the PC with a faster sampling. The microimages and the current scanning position are buffered every 2 µs, and the lifetime histogram is in a tunable integer multiple of 10 ns (Supplementary Fig. 19). The resulting raw data is, therefore, inherently temporally resolved in the microsecond regime. Through offline postprocessing, the localization itself can be performed at any integer multiple of the 2 µs microimage dwell time. Notably, during postprocessing, we are no longer forced to use fast estimators, potentially enabling more accurate measurements. The same reasoning applies to the fluorescence lifetime data.

As an example, in Fig. 3g, we display a portion of the trajectory depicted in Fig. 3f that has been rebinned at 20 µs and whose localization estimation is refined with the MLE estimator. The postprocessing pipeline not only adds a 50-fold improvement in time sampling but also reveals previously hidden spatial fast movements that were averaged out in real time due to the longer integration window.

## 4D tracking of lysosomes in living cells

To prove the versatility of our technique, we utilize it within a biological context to explore the dynamics of lysosome motion in living SK-N-BE cells. Lysosomes are membrane-enclosed organelles that play a critical role in cellular homeostasis by removing damaged organelles, as well as extraneous particles through phagocytosis[63]. Dysfunction of lysosomes has been linked to a range of diseases, including lysosomal storage disorders[64,65], neurodegenerative diseases[66,67], and cancer[68–70]. Tracking their movement is, therefore, a crucial task as it can help develop new strategies for treatments or prevention of lysosomal-related disorders[71–73]. Of particular interest are the lysosomes found at the cell periphery, which show higher mobility[74,75]. Their motion pattern is characterized by stationary states, when the lysosome is bound in one location ("stop" states), alternated with fast movements between locations ("run" states)[75–77].

To study this "stop-and-run" alternating behavior, we fluorescently labeled the lysosomal membrane with GFP by tagging the lysosome-associated membrane protein LAMP-1, and we performed 4D tracking on the organelles found at the periphery of living human neuroblastoma SK-N-BE cells. The cellular context is provided by a reference 2-dimensional image $(x, y)$ of the microtubule structure acquired right before any tracking experiment (Fig. 4a and Supplementary Video SV1). We observe that the region of the sample explored by the lysosome has a diameter of approximately 7 µm, which considerably exceeds the size of the OLV.

As expected, the lysosome displays movement along quasi-rectilinear paths, partially aligning with the microtubule structure revealed through imaging. Discrepancies between the trajectory and microtubule structure predominantly stem from the different dimensionality of the two datasets. Specifically, the recorded trajectory is in 3D space with sub-diffraction resolution (Fig. 4b and Supplementary Fig. 20), while the 2D imaging captures the projection of a single, diffraction-limited optical section. In addition, the elapsed time between the recording of the underlying image and the trajectory may contribute to these disparities.

To provide additional evidence of the role of the microtubules, we compare the motility of lysosomes in wild-type cells and after the addition of nocodazole, a drug that interferes with microtubule polymerization[78,79]. To do so, we compute the average MSD of different independent trajectories acquired before and after the treatment (Fig. 4c). The depolymerization of microtubules clearly impacts the movement of the lysosomes, which consequently exhibit a strongly subdiffusive behavior, indicating a local confinement in space (Supplementary Fig. 21).

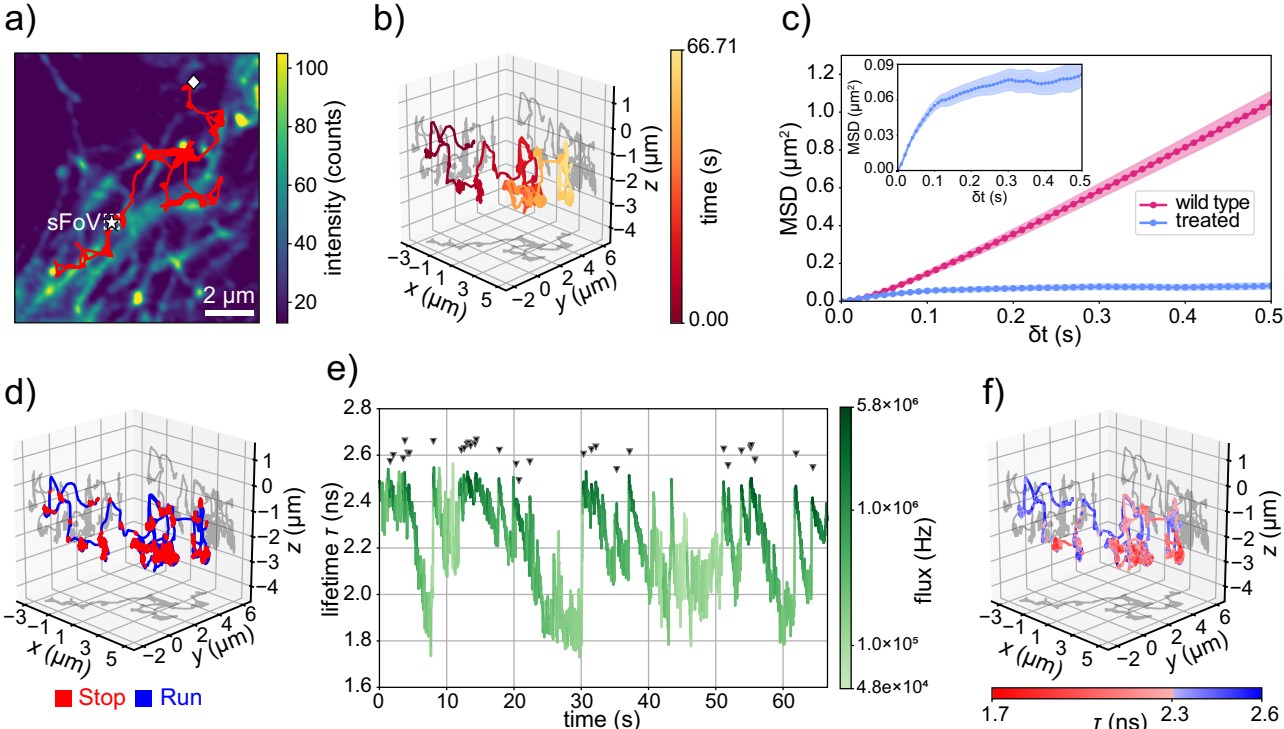

**Fig. 4 | Investigation of lysosomes diffusion with RT-4D-SPT. a** 2D projection of the spatial trajectory of a single lysosome moving inside a living neuroblastoma SK-N-BE cell. A white star and a white romb indicate the beginning and the ending of the trajectory, respectively. The Extended time trace of the trajectory is in Supplementary Fig. 20, and the video version is in Supplementary Video SV1. The organelle is tracked by exciting the GFP expressed on a membrane protein ($\lambda_{exc}$ = 488 nm) with the re-centering performed at a fixed timing $\Delta t_{rc}^{all}$ = 2.5 ms. The reference image shows the tubulin proteins labeled with Abberior LIVE 560 ($\lambda_{exc}$ = 561 nm) and is acquired prior to the lysosome tracking measurement. The pixel dwell time of 100 μs. **b** 3D plot of the trajectory shown in **a**. The colormap represents the temporal scale. **c** Mean squared displacement of lysosomes tracked in wild-type living neuroblastoma SK-N-BE cell and after the addition of nocodazole. Each curve is obtained by averaging the MSD of 15 independent single lysosome trajectories ($\Delta t_{rc}^{all}$ = 2.5 ms). The shaded area is the standard error of the mean. The mean tracking time length is 53 ± 25 s in the wild-type and 5 ± 4 s after treatment. The inset plot shows an enlarged version of the curve obtained after treatment. **d** 3D plot of the trajectory shown in (**a**). The color represents the diffusion state of the lysosome. Segmentation is performed by thresholding the average time spent by the lysosome inside voxels of dimension 5 nm × 5 nm × 20 nm (see "Methods"). **e** Time evolution of the fluorescence lifetime $\tau$ for the trajectory in (**a**) ($\Delta t_\tau$ = 10 ms). The color represents the fluorescence intensity on a logarithmic scale. The black arrows indicate when we register a rapid increment of the photon flux of at least 1 MHz. **f** 3D plot of the same trajectory of (**a**) with the color indicating the fluorescence lifetime value. Video version in Supplementary Video SV2.

We developed a simple yet effective quantitative segmentation procedure to unveil the "stop-and-run" motion pattern of the lysosomes. Briefly, the two states are differentiated by computing the residence time of the lysosome in neighboring positions inside the cell, with the "stop" states identified by long residence times. This approach is able to reveal the dual motion of the lysosomes in 3D (Fig. 4d).

When considering the full 4D dataset provided by the experiment, we can additionally analyze the time trace of the fluorescence lifetime (Fig. 4e and Supplementary Fig. 20). We measure a not-constant value of the fluorescence lifetime during the motion of the lysosome. Specifically, the fluorescence lifetime value gradually drops from the expected value for the GFP $\tau$ = 2.5 ns[80] to a plateau at 1.9 ns. Interestingly, the fluorescence photon flux varies as well between peak values of approximately 3 – 6 MHz and minima as low as 50 kHz. The 4D trajectory reveals a correlation between the two processes (Fig. 4e). By adding the fluorescence lifetime information on a 3D plot of the trajectory (Fig. 4f and Supplementary Video SV2), we observe the value of the fluorescence lifetime correlates with the motion states of the lysosome. Specifically, values around 2.4 ns are associated with the "run" state, while the gradual drop below this value is associated with the "stop" state. In the proposed case study of single lysosome tracking, the concurrent measurement of fluorescence lifetime greatly simplifies and improves the recognition of the lysosome motion state. This possibility is further confirmed in Supplementary Information Note 5, where we segment the motion state of of 15 independent 4D

lysosome trajectories by solely analyzing the fluorescence lifetime value in an automatized manner, thereby eliminating the need for any user intervention.

## Discussion

The unique nature of asynchronous read-out SPAD array detectors merges important abilities of offline and real-time SPT techniques. This class of sensors works as a camera detector – with a small sFoV and virtual sub-microsecond frame rate – and as a SPAD detector – with single-photon sensitivity and timing. By combining the SPAD array detector with a fast FPGA-based data-acquisition module, the virtual photon-counting images are used to compute in real-time the position of the particle. This position represents the input of a feedback close loop system which maintains the single particle of interest in focus – within the OLV – by driving the laser beam scanning microscope apparatus. An offline procedure can then postprocess the same virtual images for refining and rebinning the trajectory with a time resolution multiple of 2 μs. Because the real-time feedback system must simply maintain the particle within the OLV, whilst a precise particle localization can be obtained offline, the FPGA-based card can implement a fast but less accurate particle position estimation. As a result, the primary limitation of the proposed SPT approach is the actuators' lag time for re-centering the microscope focus. For this reason, we envisage a series of new RT-4D-SPT implementations in which the OLV size is changed according to the apparent diffusion coefficient of the particle. Indeed,

the excitation volume and the sFoV can be tuned by controlling the laser beam size and the overall microscope magnification on the detector plane, respectively. This scenario can be supported by the recent introduction of larger (i.e., 7 × 7 pixels) asynchronous read-out SPAD array detectors with the same temporal characteristics[48]. However, strategies to increase the size of the excitation volume should be balanced by the fact that the increase in out-of-focus fluorescence background can limit an effective increase of the OLV, leading to the same problem as in image-based SPT. Notably, the proposed combination of real-time and offline localization is also compatible with those techniques sequentially moving the beam around the particle to estimate its position (e.g., MINFLUX and orbital tracking).

A fundamental characteristic of the proposed RT-SPT approach is the ability to measure the fluorescence lifetime, opening true RT-4D-SPT experiments. The particle can be followed across a large 3D effective field-of-view, while the fluorescence lifetime can be used to understand the particle's interactions or changes in its nano-environment. Unlike current RT-SPT implementations that typically require a separate slave and dedicated DAQ for registering photon detection times, limiting the lifetime analysis to offline processing only, our RT-4D-SPT system integrates the measurement of fluorescence lifetime directly into the same DAQ and control module used for real-time particle tracking. This integration opens the possibility for intelligent experiments, where actions such as activating a second laser beam or a camera-based acquisition system can be triggered by changes in fluorescence lifetime. In summary, this architecture represents a novel advancement in the emerging field of data-driven or smart microscopy.

To demonstrate the importance of correlating dynamics information with the fluorescence lifetime, we conducted live cell experiments that monitored the diffusion behavior of lysosomes. Our findings revealed an interesting correlation between the intermittent "stop-and-run" motion pattern and the intensity and fluorescence lifetime of the GFP expressed on the organelles' membrane. The drop and recovery in intensity may be attributed to the physical 3D rolling of the organelle in the two alternating states[81], which could expose new fluorophores on the top surface. Alternatively, the reduction in intensity might result from quenching of the fluorophore by certain species when the lysosome stops. However, the variation in the fluorescence lifetime of the GFP is an unprecedented observation. Without further investigations, providing a definitive explanation for this phenomenon remains challenging. It is worth mentioning that the GFP is sensitive to various environmental parameters, including refractive index[82] and pH[83]. Thus, the change in fluorescence lifetime could signal the organelle's membrane rearrangement during its function. The lysosome experiment demonstrates the substantial amount of information potentially encoded by correlating particle movement and fluorescence lifetime. However, providing a robust biological explanation for the reason behind this correlation goes beyond the scope of this work.

In conclusion, our ability to measure both the position and the fluorescence lifetime of a particle in real time using a single instrument significantly broadens the range of biological phenomena that can be observed. Our RT-4D-SPT technique, which combines simplicity and informativeness, can potentially set a new gold standard. With the rapid development of SPAD array technologies, our approach represents a promising avenue also for single molecule tracking experiments, where the photon-flux value is substantially reduced.

## Methods
### Optical setup details
The backbone of the optical setup for the proposed RT-SPT method is a conventional laser scanning microscope, in which we replace the typical single-element detector with a 5 × 5 asynchronous read-out CMOS SPAD array detector placed in the microscope's conjugate image plane (Supplementary Fig. 22).

Laser light at the wavelength $\lambda_{exc} = 561$ nm (MPB VFL-P-1000-560) and at $\lambda_{exc} = 488$ nm (PicoQuant LDH-D-C-485) is combined and modulated in amplitude with an acousto-optic modulator (AA Opto-Electronic MT80-A1-VIS) in a dedicated laser box. Using a single-mode polarization-maintaining fiber (Thorlabs P5-405BPM-FC-2), the light is brought to the microscope setup, collimated with a reflective collimator (Thorlabs RC08FC-P01), and its polarization is cleaned and controlled with a half-wave plate (Thorlabs AHWP10M-600) and a linear polarizer beam splitter (Thorlabs CCM1-PBS251/M). The beam is reflected towards the objective at a multi-color dichroic beam splitter (Semrock Di01-R405/488/561/635-25 × 36), it passes through a first telescope formed by lenses L2 (LINOS G063-232-000) and L1 (LINOS G063-237-000) and it is scanned in the lateral directions ($x_s$ and $y_s$) via a pair of galvanometric mirrors (Thorlabs GVSM002-EC/M). A second telescope composed of the scan lens (Thorlabs SL50-CLS2) and the tube lens (Thorlabs TTL200MP) magnifies the collimated beam to an effective diameter of $\approx 46$ mm, further cropped by the usage of 1-inch optical elements. Nevertheless, the back-aperture of the 63x/1.40 oil objective (Leica HC PL APO 63x/1,40 OIL CS2) is overfilled. Finally, a quarter-wave plate (Thorlabs AQWP10M-580) turns the polarization into circular and the objective lens focuses the excitation beam into the sample. The axial position of the focal point $z_s$ is adjusted by moving the objective along the axial direction with a linear piezo-electric stage (Physik Instrumente PIFOC P-725.2CL). The sample is moved in the micrometer scale with a manual 3D microstage (Piezo-concept Manual Microstage Nikon + Piezoconcept Rectangular manual Z adjust) and moved in the nanometer scale with a 3D piezoelectric stage (Piezoconcept BIO3.300) placed on top of the microstage. The emitted fluorescence is collected in epifluorescence mode by the same objective lens, de-scanned by the galvanometric mirrors, and transmitted at the dichroic beam splitter into the detection arm. The remaining leakages of the laser illumination are blocked with two notch filters (Chroma ZET488NF and Chroma ZET561NF). The fluorescence light passes through the cylindrical lens (Thorlabs LJ1516RM-A) to induce astigmatism and it is finally focused onto the SPAD array detector by lens L3 (LINOS G063-238-000). The setup is contained in a relatively small breadboard of 800 mm × 800 mm (Standa 1B-A-80-80-015-BL) and features an overall theoretical magnification from the sample plane to the SPAD image plane of $M = 504$.

### Control module
The control module manages the microscope positioners and lasers, performing the logical operations necessary for real-time tracking. It runs a custom LabVIEW firmware and is user-controllable via a GUI.

The hardware consists of a chassis (NI PXIe-1071) that synchronizes two FPGA-based data-acquisition cards in a master/slave configuration. The master card (NI PXIe-7822R) collects single-photon pulses and computes real-time responses, while the slave card (NI PXIe-7856R) converts the master's control orders into voltage outputs. Data is sent to the host PC at specified intervals through a thunderbolt-based communication module (NI PXIe-8301) for efficient bandwidth management.

For detailed design and algorithm information, see Supplementary Information Note 6.

### Mean squared displacement calculation
Considering a set of $N_{tr}$ independent trajectories $\mathbf{r}^{(n)}(t) = \mathbf{r}_s^{(n)}(t)$ of length $L^{(n)}$ sampled with a discrete sampling step $dt_s$, we calculate the MSD as follows:

$$MSD(\delta t) = MSD(k \cdot dt_s)$$

$$= \frac{1}{N_{tr}} \sum_{n=1}^{N_{tr}} \left[ \frac{1}{L^{(n)} - k} \sum_{i=1}^{L^{(n)}-k} |\mathbf{r}^{(n)}((i+k) \cdot dt_s) - \mathbf{r}^{(n)}(i \cdot dt_s)|^2 \right]$$

It is important to note that the calculation averages over all the different trajectories and time intervals, implicitly assuming the ergodic hypothesis for the single-particle trajectories[84]. Furthermore, this formulation utilizes all the available displacements of duration $k \cdot dt_s$, increasing the averaging pool for each MSD point, but at the same time exposes the risk of correlations between overlapping displacements.

### Cell line and sample preparation

**Fluorescent beads.** To produce a sample of free diffusing fluorescent beads, we begin by diluting the beads' mother solution in distilled water. We adjust the dilution ratio between 1:200 to 1:10000 depending on bead size; larger beads require lower dilutions to obtain the same particle density in the final solution. If glycerol addition is necessary, we heat pure glycerol in a thermostatic bath at 55 °C, then add it to the bead solution until the desired volumetric glycerol/water ratio $R_{gly}$ is achieved. We vortex the solution and sonicate for 5 min. Finally, we spill a droplet of the solution directly onto a cover glass placed in the microscope's sample holder.

To produce a sample with fixed fluorescent beads, we first deposit 150 μL of poly-L-lysine (PLL) on a clean cover slip and incubate it at 37 °C for 10 min. Meanwhile, we prepare a dilution of the beads' mother solution in distilled water with a volumetric ratio between 1:500 and 1:1000, which we sonicate for 5 min. We dry the coverslip with clean air and deposit 150 μL of bead dilution on top of the adhesive film, followed by incubation at 37 °C for 10 min. We spill the remainder of the solution on the cover glass and dry it with clean air. We add 5 μL of Mowiol® mounting medium and seal the cover glass on a microscope slide.

A summary of all the beads used in the various experiments can be found in Supplementary Table 1.

**Live cells.** SK-N-BE neuroblastoma cells line are cultured in RPMI medium 1640 (Gibco), supplemented with 10% fetal bovine serum (FBS), GlutaMAX (Gibco), and penicillin/streptomycin, and induced to differentiate by 10 μM all-trans-Retinoic acid (RA, Sigma) for 5 days before imaging observation. For lysosome live monitoring, SK-N-BE cells expressing LAMP1-eGFP protein are generated. The stable cell line is obtained upon plasmid transfection (epb-bsd-EIF1a-LAMP1eGFP) using Lipofectamine™ 2000 Transfection Reagent (ThermoFisher). The cells are then selected by Blasticidin (5 μg/mL) administration. For microtubule labeling, 1 μg/mL of Tubulin Tracker™ Deep Red (Invitrogen) is added to sample media and incubated for 30 min, followed by a washout before imaging and by changing the complemented media with RPMI 1640 Medium no phenol red (Gibco), to limit signal background during imaging observation. To perturbate lysosome motility, SK-N-BE cells expressing LAMP1-eGFP are treated with 10 μg/mL nocodazole (Merck) for 1 h, which destabilizes microtubules and, as described in ref. [79], significantly increases the number of stationary lysosomes.

### Lysosome motion behavior segmentation

To differentiate between the "stop" and the "run" motion states of the lysosomes, we analyze the time spent by the organelles in neighboring positions during their movement. We start by dividing the 3D space into a regular grid with a voxel size of 5 nm × 5 nm × 20 nm. Each voxel is then associated with the number of trajectory points inside it. By multiplying with the trajectory sampling time we therefore obtain a measure of the residence time of the considered lysosome inside each voxel (Supplementary Fig. 23a). This 3D histogram is smoothed with a 3D Gaussian filter whose dimension is bigger than the voxel size to obtain the neighboring score (Supplementary Fig. 23b). This operation has the double purpose to enhance the contrast of clusters of voxels with high residence times by reducing their fluctuations and at the same time to dump the score of the "run" points, which are assumed to be mainly surrounded by empty voxels. Each trajectory point is finally

associated to a motion state by thresholding its neighboring score (Supplementary Fig. 23c, d). Depending on the particular experiment, the voxel neighboring score threshold may vary and it's up to the user to define the optimal one.

### Reporting summary

Further information on research design is available in the Nature Portfolio Reporting Summary linked to this article.

## Data availability

The experimental 3D and 4D trajectories generated and analyzed in this study have been deposited in a publicly available Zenodo repository (https://doi.org/10.5281/zenodo.11191581). However, the dataset containing the trajectories of the bead populations depicted in Fig. 3 is too large to be uploaded. Nevertheless, it is available upon request.

## Code availability

The Python data analysis source code has been deposited on a publicly available Zenodo repository (https://doi.org/10.5281/zenodo.11191581). The Lab-view data acquisition and control software used for the current study are available upon request.

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

## Acknowledgements

The authors thank all members of the Molecular Microscopy and Spectroscopy labs for the many helpful suggestions. All members of the RNA Initiative at the Istituto Italiano di Tecnologia for their contribution to the long-term vision of this project. This project has received funding from: the European Research Council, "BrightEyes", ERC-CoG No. 818699 (A.B., G.T., M.O.H., L.B., and G.V.); "ASTRA", ERC-SyG No. 855923 (I.B.); the European Union - Next Generation EU, PNRR MUR - M4C2 - Action 1.4 - Call "Potenziamento strutture di ricerca e creazione di "campioni nazionali di R&S" (CUP J33C22001130001), *National Center for Gene Therapy and Drugsbased on RNA Technology* No. CN00000041" (I.B. and G.V.); the Associazione Italiana per la Ricerca Sul Cancro, "Circular RNAs: novel players and biomarkers in tumorigenesis", IG 2019 No. 23053 (I.B.); the Ministero dell'Istruzione, dell'Università e Della ricerca (MIUR), "Non-coding RNAs, new players in gene expression regulation: studying their role in neuronal differentiation and in neurodegeneration", PRIN 2017 No. 2017P352Z4 (I.B.).

## Author contributions

G.V. conceived the idea. G.V. designed the study. I.B., E.S., and G.V. supervised the project. A.B., G.T., and M.O.H. implemented the single-particle tracking architecture. A.B., G.T., M.O.H., and L.B. implemented the data acquisition and control system. A.B. and E.S. implemented the data analysis software. E.P., F.C., and I.B. designed the live-cell experiments. A.B., E.S., M.O.H., and G.V. analyzed the data with the support of all other authors. A.B. and G.V. wrote the manuscript. All authors discussed the results and commented on the manuscript.

## Competing interests

G.V. has a personal financial interest (co-founder) in Genoa Instruments, Italy. The remaining authors declare no competing interests.
