## [Peer Review File · Nature Communications]

4D Single-Particle Tracking with Asynchronous Read-Out Single-Photon Avalanche Diode Array DetectorREVIEWERS' COMMENTS:

Reviewer #1 (Remarks to the Author):

In this report the authors present method to track freely moving particles in 3D space with very high time-resolution. Based on a confocal laser-scanning microscope that is equipped with a 5x5 SPAD array, the authors were able to asynchronously read-out the intensity distribution generated by the labelled particle and estimate its position relative to the excitation focus. By applying a FPGA based feedback algorithm, the laser-focus can be re-positioned to keep the particle in the focus. Additionally, besides the 3D position, the SPAD array detector also allows to determine the fluorescence decay-time of the particle, which can be an indicator for the local environment of the particle.

The presented method is technically relatively easy to implement also into commercial instruments and appears to be very reliable and robust. The presented experimental data very convincingly show the capabilities of the method.

The manuscript is well written, the concept is clear and sound. I found no obvious flaws. The topic perfectly fits to the scope of the journal. In summary, I recommend the publication of the manuscript.

A few corrections appear to be necessary:

p. 12, l. 39: "...show high dynamicity." I am not sure, if "dynamicity" is a valid word, but I suggest to better write "... show higher mobility" or "...show enhanced dynamics."

p. 12, l.44: "We therefore performed..." I think "therefore" does not fit well here. I suggest to either skip it, or maybe the authors wanted to use "thereafter"?

p. 13, l. 2: I suggest to use two-dimensional instead of bi-dimensional.

p. 13, l. 5: I suggest to use "...exceeds the size of the sFoV."

p. 13, l. 9: "dimensionality" instead of "dimensional"

p. 13, l. 9: "the trajectory spans through a 3D space" instead of "... spams across..."

p. 14, l. 17: "approximately 3 – 6 MHz..." instead of "approximately 3 ÷ 6 MHz..."

p. 14, l. 20: "(Fig. 4f and Supplementary Video..." instead of "(Fig. 4f ans Supplementary Video..."

Check also references 49, 56, and 60.

Reviewer #2 (Remarks to the Author):

The authors describe a real-time 3D single particle tracking system which utilizes a SPAD array for detection, combined with pulsed excitation to extract fluorescence lifetimes. This tracking method was then evaluated for its tracking capability and precision on fluorescent microspheres, and later in live cells tracking lysosomes. While the use of a SPAD array is novel in real-time tracking studies, the performance of the scope does not match or exceed existing methods in terms of tracking speed and sensitivity, nor is any new biology uncovered. The addition of fluorescence lifetime is nice, but not new and does not in and of itself require the SPAD array. I would suggest that this work is more suited for a technical journal, rather than the wide audience of Nature Communications. Even then, there are shortcomings in the current work that should preclude its publication.

Major issues:

1. The claims made regarding the tracking speed are unsupported by data. On page 9, the authors state

“Considering a minimum effective re-centering time $\Delta t_{rc} \approx 1$ ms, we can predict this optimal detection volume provides a maximum measurable diffusion coefficient for Brownian motion of $\approx 5\text{--}10 \mu\text{m}^2/\text{s}$.”

First, the re-centering time of 1 msec is not supported anywhere within the data. It is likely that the galvo mirrors can easily beat that time, but the time to recenter the objective will be slower. In fact, the authors claim (without proof) on page 12 that:

“Nevertheless, the actuation is further slowed down by the setting time of the positioner, requiring $\approx 300 \mu\text{s}$ for the galvanometric mirrors (lateral shift) and ≈ 10 ms for the objective piezoelectric stage (axial shift). The overall real-time procedure is therefore limited to a response time of around 1 ms.”

Wouldn't the response time be ultimately limited by the slowest element?

2. The above estimation is also based on the optimized detection volume (300 nm x 300 nm x 600 nm). However, this is only the detection range. The authors have not considered the effect of the size of the point-spread function (PSF). In their discussion of “Fast estimators” in the SI, they say they only considered the emission PSF. They state that when “the position approaches the edge of the field-of-view (FoV) or noise is added.” They have not considered that as the particle moves away from the center of the FoV, the emission rate will drop dramatically. It is not just the FoV that will affect the limit of detectable diffusion, but also

the size of the excitation spot. This is never discussed by the authors, nor is the size and shape of the PSF ever reported.

3. The neglected consideration of the PSF and response time informs that limit in terms of actual tracking performance. Even for fluorescent beads, which are composed of many fluorophores and extremely bright, tracking in water is never demonstrated, nor does it ever approach the supposed 5-10 $\mu\text{m}^2/\text{s}$ limit. Tracking (or trapping) at high speeds requires some sort of expanded excitation volume, usually by scanning or multiple excitation spots (see McHale, *Nano Lett.*, 2007, 7, 11,3535-3539; Hou, *Nat Comms*, 2020, 11, 3607; Cohen, *Opt. Express*, 2008, 16, 6941-6956).

4. There is an overall lack in data regarding the robustness of the tracking mechanism. For example, what is the average tracking duration when tracking at the upper limits in terms of diffusive speed? Fig. 3a has average MSD for a population of particles at different viscosities. How does the tracking time fall-off?

5. Similar to the above, what are the intensity requirements for tracking? What is the minimum intensity for successful tracking? How does this scale with diffusion coefficient? There are no intensity time traces reported for any of the free diffusion tracking. I would suspect that given the small excitation volume, there would be large dips in the intensity as the particle diffuses away from the center of the laser focus. These need to be reported.

6. The demonstration of lysosome trafficking with concurrent lifetime is interesting, however I have major concerns with how the data are interpreted. The authors state:

“Interestingly, the fluorescence photon-flux varies as well between peak values of approximately 3÷6 MHz and minima as low as 10 kHz.”

In Fig 4e, the authors show with black arrows where the intensity increases by at least 1 MHz with respect to the previous value. Again, the authors do not show the actual intensity trace. These are huge, fast changes in intensity that are likely simply “hopping” of the feedback loop onto a nearby particle. As one tracked particle bleaches to a low intensity

level, the system may lock on to a nearby brighter particle. This would be indicated by short, rapid motion and a large increase in intensity. I suspect this is the nature of the “run” states. The authors then go on to say there is a correlation between these “run” states and the fluorescence lifetime, but I suspect that this is simply a change in the fluorescence lifetime as the relative weight of the fluorescence from GFP and the background autofluorescence. The authors present this as an unprecedented observation of changes in GFP intensity and lifetime in lysosomes, but have not ruled out potential artifacts from their tracking mechanism.

Minor issues:

1. On page 6, the authors claim that “the effective achievable tracking range can easily exceed hundreds of microns.” This will not be possible along the axial direction with high NA objectives that are necessary for tracking lowly emitting targets, such as single proteins.
2. In Fig. 2h-j, the precision for XYZ localizations are reported. However, the authors change the number of photos for the Z-calculation, which gives the false impression that the precision is isotropic. Either all three should be reported for the same number of photons, or the difference in acquisition time for XY vs Z should be made clear in the figure itself. This is particularly important since the time to achieve the reported Z-precision takes 17 msec, when the claimed response time is on the order of 1 msec.
3. When calculating the CRB, the authors make the assumption that background is independent of the excitation intensity based on detector dark noise alone. This is a very poor model for real systems, particularly in cells where the autofluorescence will scale with the excitation intensity. The authors should consider adding in a real background term to the CRB calculation.
4. On page 13, the authors say that:

“The discrepancies between the trajectory and the microtubules structure is mostly attributed to the different dimensional of the two datasets, i.e., the trajectory spans across a 3D space, while the image is a single 2D optical section.”

Couldn't it be that the microtubule structure has evolved over time, or that the cell has moved? How long before the trajectory was the “reference image” collected?

Reviewer #3 (Remarks to the Author):

Bucci et al. report a new instrument and methodology, based on novel SPAD array technology, for single-particle tracking with an interesting spatio-temporal resolution combination, and the addition of simultaneous measurement of fluorescence lifetime. The paper is clearly written, the results well presented, and the main conclusions are solidly supported by the data. In my view this paper would be ready for publication after the authors address some minor issues:

Comments and Questions

- 1) “the effective achievable tracking range can easily exceed hundreds of micrometres, depending on the dynamical range of the lateral and axial positioners and the field number of the objective lens”. In principle, the sample could also be mounted on a motorized stage, to further extend the tracking range, right? Even if this is not implemented in these experiments, it may be worth mentioning.
- 2) “These assumptions are typically valid in low-excitation experiments, where fluorescence is far from saturation and the primary background source is the detector’s dark noise.” Is this the case in the experiments? Are the dark counts of the SPAD array the main contribution to background? Also, is the assumption about background independent from excitation intensity really necessary? I have the impression that it should not add much complexity to consider a background linear with excitation intensity.
- 3) In the first demonstration of 3D tracking, the authors only used the rough estimators (centroid and normalized difference). It would be nice to see how the tracking precision improves when using the MLE estimator. This could be done by off-line analysis of the registered trajectories. Also for these experiments, it would be nice if the authors mention what the total measurement time was, and discuss the influence of mechanical drift during the measurement.
- 4) Why don’t you show 4D tracking data for the 40 nm beads? Is this a signal level issue? The authors should clarify this and justify the use of the 100 nm beads
- 5) When the authors track the 100 nm beads, they observe a sub-diffusive behaviour that they ascribe to “spatial heterogeneity in the solution”. In my view this is rather unlikely. On the other hand, the authors should exclude the influence of optical forces on the larger beads, which may not be high enough to trap the bead but enough to induce a constrained

diffusion.

6) “This behavior is confirmed to a broader extent in the Supplementary Note, where we collect a pool of 15 independent 4D lysosome tracking experiments and analyze them with an automated segmentation algorithm based only on the fluorescence lifetime, hence not requiring any user intervention.” The “3” is missing after Supplementary Note. More importantly, it is not clear if the authors analyzed trajectories individually or segments of trajectories. Just before this passage, the authors showed a clear correlation between lifetime and speed (or “run” and “stop” segments). Here, it seems the authors have full trajectories of “run” or “stop” behavior. Please clarify.

Minor corrections

- “Because the information collected by the SPAD array detector is also transferred to the personal computer (PC) in the form of microimages with a high frame-rate, the spatiotemporal precision of the tracking can be improved off-line using a more precise and robust localisation algorithm than the centroid”. This phrase makes one think that the recentering is only performed in (x,y). The normalized difference algorithm for the rough axial position estimation is only introduced much later.
- There are some typographical errors to be considered (e.g. “limit” instead of “limits” in page 3, “different dimensional” in page 13)
- Figure 1a there is an error in the legend “Tube lens” and “Cylindrical lens”
- There are numerous hyperlinks in the text that, at least in my reader, lead to nowhere. Also, in page 10, there is a hyperlink instead of a reference, I think: “we calculate the MSD averaging over all the different trajectories and time intervals as described in 4.”
- “The processing pipeline is described in details in the materials and methods supplementary section 4”. There is no supplementary section 4
- Fig 3d. The authors should improve the caption and/or main text to explain what are the orange and blue curves
- What is the meaning of “The peak value is suddenly recovered on average every 3.4 ± 0.5 s”. Does the stop and go have a frequency? It seems to be a rather stochastic process.

Reviewer #1 (Remarks to the Author):

In this report the authors present method to track freely moving particles in 3D space with very high time-resolution. Based on a confocal laser-scanning microscope that is equipped with a 5x5 SPAD array, the authors were able to asynchronously read-out the intensity distribution generated by the labelled particle and estimate its position relative to the excitation focus. By applying a FPGA based feedback algorithm, the laser-focus can be re-positioned to keep the particle in the focus. Additionally, besides the 3D position, the SPAD array detector also allows to determine the fluorescence decay-time of the particle, which can be an indicator for the local environment of the particle.

The presented method is technically relatively easy to implement also into commercial instruments and appears to be very reliable and robust. The presented experimental data very convincingly show the capabilities of the method.

The manuscript is well written, the concept is clear and sound. I found no obvious flaws. The topic perfectly fits to the scope of the journal. In summary, I recommend the publication of the manuscript.

We express our gratitude to the Reviewer for recognizing the quality of our work and for emphasizing a fundamental aspect of our real-time single-particle tracking implementation – its simplicity and its potential to become mainstream due to its straightforward integration into a (confocal) laser scanning microscope. We would like to underscore that the Reviewer found our work fully aligned with the scope of Nature Communications. Finally, we would like to express our gratitude to the Reviewer for proofreading the manuscript.

A few corrections appear to be necessary:

p. 12, l. 39: "...show high dynamicity." I am not sure, if "dynamicity" is a valid word, but I suggest to better write "... show higher mobility" or "...show enhanced dynamics."

We corrected the sentence.

p. 12, l.44: "We therefore performed..." I think "therefore" does not fit well here. I suggest to either skip it, or maybe the authors wanted to use "thereafter"?

We corrected the sentence.

p. 13, l. 2: I suggest to use two-dimensional instead of bi-dimensional.

We corrected the sentence.

p. 13, l. 5: I suggest to use "...exceeds the size of the sFOV."

We corrected the sentence.

p. 13, l. 9: "dimensionality" instead of "dimensional"

We corrected the sentence.

p. 13, l. 9: "the trajectory spans through a 3D space" instead of "... spams across..."

We corrected the sentence.

p. 14, l. 17: "approximately 3 – 6 MHz..." instead of "approximately 3 ÷ 6 MHz..."

We corrected the sentence.

p. 14, l. 20: "(Fig. 4f and Supplementary Video..." instead of "(Fig. 4f ans Supplementary Video..."

We corrected the sentence.

Check also references 49, 56, and 60.

We thank the Reviewer for noticing errors in the references. Citations have been checked and the books corrected.

Reviewer #2 (Remarks to the Author):

The authors describe a real-time 3D single particle tracking system which utilizes a SPAD array for detection, combined with pulsed excitation to extract fluorescence lifetimes. This tracking method was then evaluated for its tracking capability and precision on fluorescent microspheres, and later in live cells tracking lysosomes. While the use of a SPAD array is novel in real-time tracking studies, the performance of the scope does not match or exceed existing methods in terms of tracking speed and sensitivity, nor is any new biology uncovered. The addition of fluorescence lifetime is nice, but not new and does not in and of itself require the SPAD array. I would suggest that this work is more suited for a technical journal, rather than the wide audience of Nature Communications. Even then, there are shortcomings in the current work that should preclude its publication.

We sincerely appreciate the Reviewer for their insightful comments and valuable suggestions, which have significantly contributed to enhancing the clarity and rigor of our manuscript. In addressing the major criticism raised by the Reviewer regarding the perceived lack of substantial advancement over existing methods, we will first provide a comprehensive response before delving into a point-by-point reply to the comments and requests.

The real-time single-particle tracking (RT-SPT) method proposed in our study leverages an asynchronous read-out single-photon avalanche diode (SPAD) detector array. This technology, characterized by exceptional temporal resolution limited only by the tens of nanoseconds dead-time of the individual pixels, imposes no practical constraints on the spatiotemporal resolution of the tracking experiment. Our RT-SPT spatiotemporal resolution is contingent solely upon the emitter's photon flux, essentially the signal-to-noise ratio, and the lag time of the actuators for re-centering the excitation spot—standard considerations for any RT-SPT method. This innovative aspect holds promise for opening new avenues in the field. Moreover, its seamless implementation into a conventional confocal laser-scanning microscope (CLSM) by replacing the pinhole and single-element detector with a SPAD array detector is groundbreaking. Major microscopy instrument manufacturers such as Genoa Instruments, Abberior Instruments, Nikon Instruments, and Picoquant have already adopted this detector in their commercial CLSM, with Zeiss, Leica, and ISS exploring similar possibilities. This facilitates the widespread dissemination of our RT-SPT approach, an achievement not yet realized by any other RT-SPT, including the orbital-tracking approach, currently considered the most straightforward RT-SPT implementation. Similar to our RT-SPT, the orbital-tracking approach does not require substantial changes to the CLSM architecture, such as altering the scanning apparatus to enable a fast "image" of the volume around the particle to track—or, more generally, the particle localization.

Crucially, our implementation's performance is comparable to that of existing methods, including the orbital-tracking approach and other RT-SPT methods based on Gaussian excitation spot. Moreover, our RT-SPT principle is compatible with MINFLUX single-particle tracking (based on donut excitation spot) recognized for establishing the ultimate spatiotemporal resolution in single-molecule tracking. Our published theoretical work (Slenders E., Phys. Rev. Research 5, 023033, 2023) supported by ongoing experimental results, already showcases the synergistic advantages of combining a SPAD array detector with MINFLUX for imaging. The logical progression would be to extend MINFLUX for tracking with a SPAD array detector, representing the next natural step in our research trajectory.

Additionally, we wish to underscore an aspect that the Reviewer may not have fully considered—the implementation of fluorescence lifetime estimation in our study. Our approach stands out as the first to eliminate the requirement for a separate slave device, which impose an offline analysis of fluorescence lifetime. Instead, it seamlessly integrates into the same data-acquisition and controlling board (an FPGA-based board) responsible for real-time feedback systems. This innovation holds immense promise for real-time event-driven operations, such as activating a second probing laser based on fluorescence lifetime,

presenting exciting prospects for dynamic experiments. To better highlight this point we add this sentence in the Discussion Section:

“Unlike current RT-SPT implementations that typically require a separate slave and dedicated DAQ for registering photon arrival times, limiting the lifetime analysis to offline processing only, our RT-4D-SPT system integrates the measurement of fluorescence lifetime directly into the same DAQ and control module used for real-time particle tracking. This integration opens the possibility for intelligent experiments, where actions such as activating a second laser beam, or a camera-based acquisition system can be triggered by changes in fluorescence lifetime. In summary, this architecture represents a novel advancement in the emerging field of data-driven or smart microscopy.”

We have elaborated on these pivotal points in the revised manuscript, highlighting in different part of the manuscript the unique contributions of our approach in pushing the boundaries of real-time single-particle tracking. We trust that these clarifications will address the concerns raised and underscore the significance of our work in advancing the field.

Major issues:

1. The claims made regarding the tracking speed are unsupported by data. On page 9, the authors state “Considering a minimum effective re-centering time $\Delta t_{rc} \approx 1$ ms, we can predict this optimal detection volume provides a maximum measurable diffusion coefficient for Brownian motion of $\approx 5\text{--}10 \mu\text{m}^2/\text{s}$.” First, the re-centering time of 1 msec is not supported anywhere within the data. It is likely that the galvo mirrors can easily beat that time, but the time to recenter the objective will be slower. In fact, the authors claim (without proof) on page 12 that:

“Nevertheless, the actuation is further slowed down by the setting time of the positioner, requiring $\approx 300 \mu\text{s}$ for the galvanometric mirrors (lateral shift) and ≈ 10 ms for the objective piezoelectric stage (axial shift). The overall real-time procedure is therefore limited to a response time of around 1 ms.” Wouldn't the response time be ultimately limited by the slowest element?

We agree with the Reviewer's observation that the information relative to the maximum measurable diffusion coefficient reported in the main text is inaccurate. Furthermore, we acknowledge that the overall concept is inadequately described and, to some extent, lacks comprehensive support from the available data. In response to this feedback, we have added the Supplementary Note 3 entirely dedicated to this aspect in which we discuss more coherently the parameters involved and the characterization of the actuators' speed. For improving readability, now the main text of the paper simply reports the results:

“Considering the rise time of the positioners ($\approx 200 \mu\text{s}$ laterally and ≈ 4 ms axially) together with the dimension of the OLV, we predict a maximum measurable diffusion coefficient for Brownian motion of $\approx 10 \mu\text{m}^2/\text{s}$ (Supplementary SI Note 3). This upper limit matches with the diffusion coefficients of relatively small proteins (above 30 – 40 kDa) moving in the cytoplasm and the membrane of cells [55–57], while it may be unsuitable for diffusion of the same protein in pure water [58].”

and links to the Supplementary Note 3 for details.

Here, we present a schematic and concise overview of the arguments used to evaluate the 3D maximum measurable diffusion coefficient.

1. Determination of the re-centering time for lateral and axial directions.
 - a. We performed a direct measurement of the actuators' response to a step-like input (see Rebuttal Figure 1)

- b. The re-centering time is set equal to the rise time, which is defined as the time interval that the actuator needs to move from 10% to 90% of the target value. Rise time values are given for each dimension in the caption.
 2. From the calculated re-centering times we calculated the value for lateral and axial diffusion coefficient, respectively. Here we used the Einstein formula for the Brownian motion: $\langle dx^2 \rangle = 2 \cdot n \cdot D \cdot dt$
 - a. We set dt to the rise time of the respective actuator and dx as the half-width of the optimal detection volume in the corresponding direction (x , y , or z). The dimensionality n is set to 1.
 - b. Thus, we get $D_x = 61 \text{ } \mu\text{m}^2/\text{s}$, $D_y = 65 \text{ } \mu\text{m}^2/\text{s}$, $D_z = 9.6 \text{ } \mu\text{m}^2/\text{s}$
 3. For 3D tracking, we set the maximal trackable diffusion equal to the lowest value of trackable diffusion:
 - a. Thus: $D_{3D} = 10 \text{ } \mu\text{m}^2/\text{s}$

Rebuttal Figure 1 - Rise time of the positioners. a) Comparison between the input signal (red) and the corresponding positioner response (blue). All readings are taken from the positioner driver's sensor output port, except for the z-axis input signal which is taken directly from the FPGA. The displacement imposed in each direction matches with the half-width of the OLV. The response is fitted with a logistic function (black) with the dashed lines marking the 10 % and 90 % levels from which the rise time t_r is calculated. The curves in x are the average of 898 cycles and yield $t_x = 185 \pm 2 \text{ } \mu\text{s}$. The curves in y are the average of 897 cycles and yield $t_y = 174 \pm 2 \text{ } \mu\text{s}$. The curves in z are the average of 597 cycles and yield $t_z = 4.71 \pm 0.03 \text{ ms}$. Adapted from SI Note Figure SN7

2. The above estimation is also based on the optimized detection volume (300 nm x 300 nm x 600 nm). However, this is only the detection range. The authors have not considered the effect of the size of the point-spread function (PSF). In their discussion of "Fast estimators" in the SI, they say they only considered the emission PSF. They state that when "the position approaches the edge of the field-of-view (FoV) or noise is added." They have not considered that as the particle moves away from the center of the FoV, the emission rate will drop dramatically. It is not just the FoV that will affect the limit of detectable diffusion, but also the size of the excitation spot. This is never discussed by the authors, nor is the size and shape of the PSF ever reported.

We agree with the Reviewer's observation that the optimal detection volume (OLV) does not coincide with the field of view of the detector array, referred to as the static field-of-view (sFoV), but rather depends on various parameters, including — but not limited to — the size of the excitation point-spread-function (PSF) of the system. Throughout our work, we have consistently considered this aspect. For instance, the OLV is derived from actual experiments in which a particle is moved across the sFoV, utilizing a dedicated and precise xy piezo stage, and localized multiple times. Each localization is performed with a fixed acquisition time and therefore naturally accounts for the drop in excitation intensity as the particle moves away from the center of the sFoV.

We realize more clarity is necessary to highlight this aspect. Therefore, we have added key paragraphs in the Main Text and explicitly reported the intensity of the simulated micro-images in the main figures. Furthermore, the derivation of the Cramér-Rao bound (CRB) in the Supplementary Note 1 has been deeply revised, with a clear definition of the PSF and a relative experimental characterization (SI Note Figure SN1).

In the following we want to shortly clarify the approach we have used in our work, which is reflected both in experiments and simulations.

In the main text, in the Section “Localization within the static field-of-view” (page 7), we introduced the parameter $N(\mathbf{r}_e)$, which is the total number of photons detected by the sensor for a fixed acquisition time. This parameter explicitly depends on the position of the particle relative to the center of the sFoV, referred to \mathbf{r}_e , and it is defined in the Supplementary Equation S.13. Briefly, once the particle moves away from the center, this parameter decreases due to:

- (1) the reduced excitation intensity at its position (based on the spatial shape of the excitation) and
- (2) the reduced overlap of the emission PSF of the particle and the physical size of the detector (the “cropping” effect).

Furthermore, a space-independent background is also added (Supplementary Note Equation S.7).

Consequently, also the signal-to-background ratio depends on the emitter position \mathbf{r}_e .

The calculations of the CRB shown in Figure 2 (a) and (d) render exactly this tendency, by answering the question: What is the localization uncertainty of a particle at position \mathbf{r}_e given the condition that from the same particle would be detected N_p photons with an SBR_p if it was in the center of the sFoV? The uncertainty in the periphery increases mainly due to the lower number of detected photons from the emitter (see Supplementary Note Figure SN2).

Regarding the fast estimators, and in particular the centroid, we would like to elaborate on how the variation of the number of detected photons has been addressed. Supplementary Note Equation S.17 describes the estimated emitter position in presence of background. For the reasons explained above, the $SBR=SBR(\mathbf{r}_e)$ is space dependent and brings into the formula the information about the total number of detected photons. Even without the “cropping” effect (which means assuming an infinitely big detector), we would still get a non-linearity due to the diffraction-limited excitation causing a variation of the signal (hence SBR) thorough the sFoV. We have added this case scenario in the simulation of Rebuttal Figure 2 which also helps visualize what happens when both “cropping” and diffraction-limited illumination are considered.

The following corrections/changes in the Main Text has been introduced:

“As described before, in RT-SPT, the ability to track a single moving particle depends on the precise, accurate, and timely localization of its position. This enables the re-centering of the excitation volume – and thus the sFoV – on the particle before the particle leaves the sensitive region. Consequently, the characterization of the localization performance stands as a critical first step in thoroughly assessing our RT-SPT performances. In particular, it is crucial to define the region in which – for a given signal-to-background ratio – the particle can be reliably localized, which we named the optimal localization volume (OLV). Evidently, the OLV does not necessarily coincide with the sFoV as it is also influenced by the dimension and shape of the excitation volume: the photon flux from the particle decreases moving away from the center of the Gaussian excitation volume, leading to a decrease in localization precision.”

Rebuttal Figure 2 - Simulated linearity of the centroid estimator as a function of the static field-of-view and the signal-to-noise ratio. Simulated centroid estimation of a lateral shift for different combinations of sFoV size and SBR level. Note how the background level is characterized by its peak value $SBRp=SBR(r_e=0)$. Adapted from SI Note Figure SN4

3. The neglected consideration of the PSF and response time informs that limit in terms of actual tracking performance. Even for fluorescent beads, which are composed of many fluorophores and extremely bright, tracking in water is never demonstrated, nor does it ever approach the supposed $5-10 \mu\text{m}^2/\text{s}$ limit. Tracking (or trapping) at high speeds requires some sort of expanded excitation volume, usually by scanning or multiple excitation spots (see McHale, Nano Lett., 2007, 7, 11,3535-3539; Hou, Nat Comms, 2020, 11, 3607; Cohen, Opt. Express, 2008, 16, 6941-6956).

We trust that our responses to the first and second Reviewer's concerns have successfully conveyed that our RT-SPT approach considers the size of the excitation and emission Point Spread Functions (PSFs). While this aspect may not have been adequately clarified in the original version, we have made substantial changes to the main text and added a Supplementary Note (SI Note 1) that thoroughly describes this critical dependency. Furthermore, these responses also provide a critical analysis, supported by new experiments, of the response time of all actuators used to implement real-time tracking, concluding that for the given optimal localization volume of our system, $10 \mu\text{m}^2/\text{s}$ is a realistic tracking limit of our implementation. This conclusion is fully detailed in the new Supplementary Note (SI Note 1).

Motivated by the new analysis described above and in response to the Reviewer's request, we conducted a new set of experiments aimed at tracking particles of different sizes (40 nm, 100 nm, and 200 nm diameter) freely diffusing in water. The results are summarized in the Rebuttal Figure 3 and presented in the main text in a new Figure 3 and various Supplementary Figures (Supplementary Figure S5, S6, S7). These experiments confirmed our implementation's capability to track particles up to $10 \mu\text{m}^2/\text{s}$. Specifically, we report a trajectory of a 40 nm bead emitting up to 500 KHz photons and moving at $11 \mu\text{m}^2/\text{s}$ for more than a second.

This set of new experiments also underscores the high-throughput capability of our implementation. By utilizing appropriate intensity threshold values to initiate and stop tracking, multiple particle traces (hundreds of particles) can be obtained in just a few minutes.

Rebuttal Figure 3 - 4D tracking on free fluorescent beads in water. **a)** Distributions of the measured diffusion coefficients for different fluorescent beads ($\lambda_{\text{exc}} = 488$ nm) freely diffusing in pure water. The legend reports the number of single trajectories acquired per bead size. For each single bead trajectory, the re-centering in the lateral and axial directions is performed at a fixed timing of $\Delta t^{\text{lat}} = 1$ ms and $\Delta t^{\text{ax}} = 2$ ms respectively. **b)** Distributions of the fluorescence lifetimes for the same fluorescent beads populations in a. **c)** Example of the MSD of three single trajectories from the beads populations in a. Each curve is fitted with a linear model and the relative diffusion coefficient D is displayed accordingly. The legend reports the time length T_L of each trajectory. Adapted from Figure 3

Finally, we would like to address the Reviewer's comment regarding the necessity of expanding the static field of view (sFOV) of the system for tracking "fast" particles. We concur with the Reviewer that enlarging the sFOV is a valuable approach for tracking fast particles. As an example, we consider the 3D-SMART real-time single-particle tracking (SPT) implementation from the Welsher group, which rapidly scans a beam across a $1 \times 2 \mu\text{m}^3$ volume to enhance the sFOV, currently recognized as one of the best architectures for fast tracking. However, it's important to note that this approach also comes with certain limitations.

Scanning fast (tens of microseconds) this relatively large volume with a high-numerical aperture objective requires the use of a pair of electro-optical deflectors (EODs) and a tunable acoustic gradient (TAG) lens, which adds complexity to the system. In this context, it is important to highlight that, although introduced in 2019, asynchronous read-out single-photon avalanche diode (SPAD) arrays are already becoming the detector of choice for confocal laser-scanning microscopy. Major brands such as Nikon, Zeiss, Abberior, and Picoquant, as well as smaller companies like Genoa Instruments and ISS, are already offering confocal microscopy equipped with these or similar array detectors. Furthermore, the use of EODs precludes a descanned detection and thus the use of a pinhole, which is fundamental when tracking on a complex sample (such as within a cell or a tissue) to reduce the out-of-focus background and achieve a good signal-to-background ratio. Notably, our de-scanned architecture and our SPAD array detector allow for a 1.4 Airy unit pinhole confocality. Finally, the large static field of view (sFOV) may necessitate working at lower particle concentrations to avoid the presence of more than one particle in the sFOV.

While real-time single-particle tracking (RT-SPT) techniques such as 3D-SMART adopt strategies to enlarge the static field of view (sFOV) and thus the optimal localization volume (OLV) to improve spatiotemporal resolution, techniques such as MINFLUX employ the opposite strategy. MINFLUX effectively reduces the OLV by utilizing a doughnut-shaped excitation volume (the localization precision of MINFLUX outside the orbital trajectory, typically on the order of tens of nanometers in diameter, rapidly diverges). This approach allows for much higher precision with a reduced number of photons. Coupled with fast electro-optical deflectors (EODs) to implement OLV repositioning, MINFLUX achieves extraordinary spatiotemporal resolution.

These observations about 3D-SMART and MINFLUX suggest that these implementations, along with our SPAD array-based approach, can pave the way for new implementations that merge the best innovations from each.

4. There is an overall lack in data regarding the robustness of the tracking mechanism. For example, what is the average tracking duration when tracking at the upper limits in terms of diffusive speed? Fig. 3a has average MSD for a population of particles at different viscosities. How does the tracking time fall-off?

The series of experiments tracking fluorescent beads of various sizes in water, introduced in response to the previous Reviewer's question, provides a comprehensive answer to the current query as well. In Rebuttal Figure 4a we present the histogram of tracking durations for beads of different sizes while rebuttal Figure 4b shows the scatter plot of the tracking duration against the diffusion coefficient. As correctly anticipated by the Reviewer, the tracking duration does fall-off for faster diffusion. This behavior is further confirmed when the same analysis is performed on the beads diffusing in water-glycerol solutions (Supplementary Figure 4c,d).

Rebuttal Figure 4 - Tracking length characterization. a) Distributions of the measured diffusion coefficients for the beads populations of Figure 3a,b and Supplementary Figure S4. Adapted from Supplementary Figure S5 b) Scatter plot of the tracking length vs diffusion coefficient for the same fluorescent beads populations in a. d) Distributions of the measured diffusion coefficients for the water-glycerol dilutions of Supplementary Note Figure SN8. d) Scatter plot of the tracking length vs diffusion coefficient for the same fluorescent beads populations in c.

Regarding the absolute value of the tracking duration, it is nevertheless crucial to emphasize that these high-throughput experiments were purposefully designed to gather a statistically significant dataset for estimating the diffusion coefficient. We prioritized the number of acquired trajectories over extended tracking durations. To address the concern about the capability for prolonged tracking, we successfully optimized the tracking parameters for extended tracking periods, enabling us to follow a 200 nm bead emitting 10 MHz and diffusing at $0.5 \mu\text{m}^2/\text{s}$ for more than 2 minutes (Figure 3e,f and Supplementary Figures S6 and S7). This dual demonstration highlights the versatility of our approach, showcasing both the ability to provide comprehensive diffusion coefficient data and the capacity for extended tracking when desired.

To further demonstrate the tracking length capabilities of our RT-SPT technique, we repeated of the “controlled” single-particle tracking experiment (Figure 2g,h and Supplementary Figure S3). Specifically, we moved a 20 nm fluorescent bead along a predefined 3D trajectory using a 3D piezo stage, and we employed our system to track the bead. By optimizing the excitation intensity to achieve a 50 KHz photon-flux, we successfully tracked the bead, which moved at a constant tangent velocity of $5.5 \mu\text{m}/\text{s}$, for more than 25 seconds with minimal photobleaching. Notably, in this experiment, the repositioning time was not predetermined but automatically updated—both laterally and axially—by the system to maintain 100 photons per repositioning event. Given that the beads emitted at 50 KHz, the average repositioning time was approximately 2 ms. This experiment highlights the asynchronous read-out aspect of the SPAD array, indicating the absence of a minimum frame rate as seen in conventional cameras. Thus limiting the

spatiotemporal resolution of our system solely to the particle photon flux and the repositioning time of the actuators.

5. Similar to the above, what are the intensity requirements for tracking? What is the minimum intensity for successful tracking? How does this scale with diffusion coefficient? There are no intensity time traces reported for any of the free diffusion tracking. I would suspect that given the small excitation volume, there would be large dips in the intensity as the particle diffuses away from the center of the laser focus. These need to be reported.

In response to the reviewer's suggestion, we have added time traces of all trajectories presented in the paper to the Supplementary Material. Specifically, we have included traces showing the particle position, intensity, and, where available, fluorescence lifetime (Supplementary Figures S3, S5, S6, S8).

We acknowledge the importance of intensity requirements as fundamental metrics for RT-SPT techniques, and we have taken steps to address this in our manuscript. Specifically, we have modified both the main text and the Supplementary material to underscore any time the intensity level plays an important role, the most important one being in the re-centering condition. To summarize, our tracking system continuously monitors the particle until its intensity falls below a user-defined threshold, as depicted in the flowchart in Supplementary Figure S13. This decrease in intensity can occur for three primary reasons:

- (1) Particle bleaching.
- (2) The particle moving inside the optimal localization volume, but localized with a localization uncertainty so high that the system re-centers the sFoV incorrectly, resulting in a drop in intensity.
- (3) Rapid diffusion during the localization time window, causing the particle to evade the optimal localization volume and hence reproducing the scenario described in point (2).

While reducing the emitted photon flux can mitigate the effect outlined in point (1), this solution conflicts with the implications of points (2) and (3). Notably, if uncertainty is the limiting factor, as described in point (2), then increasing the signal-to-background ratio by boosting the photon flux becomes necessary. Similarly, if rapid particle movement is the issue, shortening the localization window is required, demanding an increased photon flux to collect the same amount of photons in a shorter duration.

Therefore, it's evident that producing long trajectories requires the user to know the sample and experimental conditions, finding an optimal balance point between these effects by adjusting excitation intensity and tracking thresholds. This optimization problem has been already anticipated in the previous answer (Reviewer's question 4), where the acquisition of the trajectories of 200 nm fluorescent beads was tailored either for statistical ensemble analysis or for single prolonged trajectories.

Despite the complexity of this argument, we aim to establish a reasonable minimum photon flux necessary for successful tracking. We begin by assuming an SBR of 5 to be a realistic condition, as already employed in Figure 2 and Supplementary Figure S2. Lower values could notably compromise the linearity and uncertainty of fast estimators. Given that our SPAD array registers approximately 3 kHz of dark count rates (as illustrated in Rebuttal Figure 5), we calculate an overall photon flux (signal + dark) of 18 kHz. To maintain the same level of uncertainty as depicted in Figure 2, we must collect at least 100 photons, a task achievable in approximately 5 ms with the aforementioned photon flux. This temporal constraint matches with the setting time of the slowest actuator, as previously utilized to estimate the maximum measurable diffusion coefficient. Consequently, an emission rate of 20 kHz should suffice to ensure proper tracking up to $10 \mu\text{m}^2/\text{s}$.

Dark counts (Hz) - Total = 2635 Hz

92	55	236	58	374
56	56	57	59	57
644	61	55	55	140
66	59	53	57	59
63	55	54	54	58

Rebuttal Figure 5 – Dark counts characterization. Average map of the dark count rates for the 25 elements of the SPAD array detector (values in Hz). The result is the average of 100 acquisitions obtained by completely covering the detector and counting the (fake) photons for 1 s.

However, it's important to note that considering the background composed only of dark counts is optimistic, as real scenarios most likely will result in increased background due to reflection and out-of-focus fluorescence. For instance, the solutions containing freely diffusing beads exhibited an augmented background of at least 5-6 kHz, thereby increasing our photon flux requirement up to 30-35 kHz. In fact, we observe a negligible number of trajectories acquired with less than 50 kHz of photon flux, irrespective of bead size (as illustrated in Rebuttal Figure 6a).

Rebuttal Figure 6 - Tracking intensity characterization. Distributions of the photon fluxes for the beads populations of Figure 3a,b and Supplementary Figure S5. Adapted from Supplementary Figure S4

6. The demonstration of lysosome trafficking with concurrent lifetime is interesting, however I have major concerns with how the data are interpreted. The authors state: "Interestingly, the fluorescence photon-flux varies as well between peak values of approximately 3÷6 MHz and minima as low as 10 kHz." In Fig 4e, the authors show with black arrows where the intensity increases by at least 1 MHz with respect to the previous value. Again, the authors do not show the actual intensity trace. These are huge, fast changes in intensity that are likely simply "hopping" of the feedback loop onto a nearby particle. As one tracked particle bleaches to a low intensity level, the system may lock on to a nearby brighter particle. This would be indicated by short, rapid motion and a large increase in intensity. I suspect this is the nature of the "run" states. The authors then go on to say there is a correlation between these "run" states and the fluorescence lifetime, but I suspect that this is simply a change in the fluorescence lifetime as the relative weight of the fluorescence from GFP and the background autofluorescence. The authors present this as an unprecedented observation of changes in GFP intensity and lifetime in lysosomes, but have not ruled out potential artifacts from their tracking mechanism.

We appreciate that the Reviewer found the prospect of correlating particle dynamics information with fluorescence lifetime intriguing. Indeed, tracking the particle while utilizing the fluorescence lifetime to unveil the nano-environment conditions of the particle and its interactions could yield new insights into various biological processes. In this study, we harness the well-documented 'stop-and-run' behavior of lysosomes to showcase the effectiveness of our proposed RT-SPT in tracking particles within a biological context, particularly within a living cell. Regarding the Lysosome experiments, we agree with the Reviewer that more validations are required to attribute a biological meaning to the correlation observed between lifetime and the 'stop' and 'run' phases; however, this is beyond the scope of this work. On the other side, we agree with the Reviewer on the importance to verify the absence of artefacts in the tracking experiments. Specifically:

- (1) "Hopping" denoting the possibility of the system losing track of the particle and mistakenly initiating tracking of a nearby particle while assuming a single trace;
- (2) Incorrect lifetime estimation, stemming from potential issues such as the pile-up effect, uncorrelated background, or a short-lifetime background (e.g., autofluorescence or scattering).

The (1) "Hopping" issue is inherent in any single-particle tracking technique, whether camera-based or real-time, and is challenging to eliminate entirely but can be mitigated. For our RT-SPT experiments, we implemented various strategies. The primary approach to minimize this problem involved reducing the particle concentration, thereby decreasing the likelihood of two particles simultaneously occupying the static field-of-view (sFOV) of the system. It's crucial to note that the sFOV of our system is well below $1 \mu\text{m}^3$, a volume that further diminishes when considering the optimal localization volume. In our experiments, we initiated the tracking in the peripheral region of the cell, where particle density is lower than in the perinucleus (Rebuttal Figure 7).

Another strategy to reduce the probability of "Hopping" is to properly set the intensity threshold below which the tracking stops. In the Lysosomes tracking experiments we used a threshold of 50 kHz well above the 10 kHz overall background) that we measured in the cell in absence of particle in the sFOV. As mentioned in the previous answer, this value accounts for both the dark-noise of the detector (3 KHz) and the autofluorescence, cross-talk Tubulin labeling, scattering, etc. This setting should automatically stop the tracking when the sFOV does not contain any particle, before a new one is randomly found.

Rebuttal Figure 7 - Tracking intensity characterization. Images of the lysosome distribution in the perinuclear region of SK-N-BE cells. Image a is acquired with 100 μs pixel dwell time. Image b is acquired with 200 μs pixel dwell time.

In addressing the issue (2) of incorrect fluorescence lifetime estimation, the range of photon flux observed during the tracking experiments (50 kHz – 5 MHz) is incompatible with certain potential sources of error and mitigate others. Specifically, a photon flux below 5 MHz prevents the saturation of the time-correlating single-photon counting measurement that could otherwise result in an underestimation of the lifetime value due to the pile-up effect. In a prior study conducted by our group (Tortarolo et al., *Advanced Photonics*, Vol. 6, Issue 1, 016003, 2024), we characterized a 3% error in lifetime estimation only when the photon flux exceeded 70 MHz, attributing it to the pile-up effect. Furthermore, maintaining a photon flux above 50 kHz ensures the mitigation of background contribution in the lifetime measurement. It's important to note that a high uncorrelated background may result in a longer lifetime estimation, while elevated auto-fluorescence and scattering backgrounds may lead to a shorter lifetime.

To explore these potential contributions to the lifetime estimation, we correlated the photon flux with the fluorescence lifetime for two representative Lysosome traces acquired before and after the addition of nocodazole, a drug that interferes with microtubule polymerization (Rebuttal Figure 8).

Rebuttal Figure 8 – Correlation between photon flux and fluorescence lifetime in lysosomal trajectories. a) Trajectory acquired in wild type SK-N-BE live cell. b) Trajectory acquired after the addition of nocodazole

Notably, all the experimental conditions between the two sample were maintained the same. While the graph analyzing the trace in wild type (Rebuttal Figure 8a) reveals a slight correlation between the photon flux and the lifetime (higher photon flux corresponds to a longer lifetime), the trace in the absence of microtubules does not exhibit such a correlation (Rebuttal Figure 8b). Therefore, this analysis partially dismisses the possibility that changes in lifetime are predominantly due to a background-induced artifact. Instead, it

suggests the necessity of a more intricate explanation for the observed changes in lifetime, potentially involving Lysosome functions.

Considering the discussion and analysis provided above, we would like to address the Reviewer's concern suggesting that the alterations in the lifetime analysis are solely attributed to a combination of the hopping effect and background-induced artifacts.

As per the Reviewer's request, we have introduced a new Supplementary Figure S8, displaying comprehensive 4D data traces that include the intensity trace (photon-flux), position across the three spatial coordinates, and fluorescence lifetime, all as a function of time. Upon examination, there appears to be some resemblance between the lifetime and photon flux time traces. As emphasized in the main text of the paper, in some cases, the recovery of the lifetime and a peak of intensity seem to coincide, but not consistently across all cases. Furthermore, the decrease in both quantities is not entirely correlated, as many intensity dips don't align with lifetime drops. This lack of correlation indicates that the reduction in lifetime is not solely a consequence of the signal-to-background reduction. These initial observations suggest that changes in fluorescence lifetime cannot be explained solely by potential "hopping" artifacts but must be analyzed in the context of the well-known stop-and-run behavior of Lysosomes. We appreciate the Reviewer's suggestion for the full visualization of the 4D temporal traces, as it significantly aids in interpreting the data. Supplementary Figure S14 also provides additional evidence that the "stop-and-run" behavior is not a complete misinterpretation of the "hopping" artifact. Given that a lysosome is always contained in the sFoV region, if we assume that there is no physical "run" state this artifact could still be caused by the system "hopping" from one Lysosome to another. Observing the 2D (for simplicity) trajectory, as taken from the Supplementary Figure S8, we nevertheless realize this is not possible. In fact we recall the sFoV is around 650 nm in the sample space and therefore does almost never contains two "stop" regions, which would be necessary for swapping between stationary lysosomes (Rebuttal Figure 9).

Rebuttal Figure 9 – 2D projection of the final segmented trajectory. Segmentation of the trajectory in Rebuttal Figure 8a. Adapted from Supplementary Figure S14.

Additionally, from Supplementary Video SV1 some paths are traveled back and forth, which would be hard to justify in the hopping hypothesis.

We trust that this discussion persuades the reviewer that the correlation between fluorescence lifetime and the type of motion observed in Lysosomes is not a mere artifact; rather, its origin must be sought in the underlying bio-mechanisms. However, we acknowledge that this preliminary observation cannot provide conclusive evidence, and further experiments are warranted for a comprehensive understanding of this phenomenon. It is essential to reiterate that delving deeper into this aspect is beyond the scope of the current work.

Minor issues:

1. On page 6, the authors claim that “the effective achievable tracking range can easily exceed hundreds of microns.” This will not be possible along the axial direction with high NA objectives that are necessary for tracking lowly emitting targets, such as single proteins.

We acknowledge the Reviewer's observation that the effective achievable axial tracking range is inherently restricted by the working distance of the objective lens, which is approximately 150 μm for our objective lens (Leica HC PL APO 63x/1,40 OIL CS2). In the lateral direction, the tracking range can potentially be higher. However, it is essential to note that in both cases, aberrations may introduce biases in the estimation process: field curvature in the lateral direction and spherical aberrations in the axial direction. We have addressed and clarified this concern in the main text:

In the Introduction Section: *“The detection scheme enables direct and almost instantaneous 3D localization of particles within a relatively small volume, in the order of the size of the sub-micrometer focal excitation volume, in all directions (x, y, z). This information enables dynamic repositioning of the beam scanning system to keep the particle centered in the excitation volume. As a result, the effective tracking range of the real-time single-particle tracking is primarily constrained by the lateral and axial scanning capabilities of the microscope.”*

In the Result Section: *“Regarding the spatial range of our RT-4D-SPT method, while the sFoV is confined to a few hundred nanometers, the effective tracking range can vary significantly, spanning orders of magnitude depending on the effective scanning capabilities of the microscope. In the case of high numerical objective lenses such as our implementation, these ranges can extend over a few hundred micrometers laterally and around a hundred micrometers axially. However, it's crucial to acknowledge that optical aberrations, encompassing field curvature and spherical aberration along the lateral and axial directions, respectively, may substantially reduce these practical tracking values.”*

2. In Fig. 2h-j, the precision for XYZ localizations are reported. However, the authors change the number of photons for the Z-calculation, which gives the false impression that the precision is isotropic. Either all three should be reported for the same number of photons, or the difference in acquisition time for XY vs Z should be made clear in the figure itself. This is particularly important since the time to achieve the reported Z-precision takes 17 msec, when the claimed response time is on the order of 1 msec.

We acknowledge the Reviewer's concern regarding potential confusion arising from the use of different numbers of photons for lateral and axial repositioning in the induced RT-SPT experiment. To address this, we have repeated the experiment with the same number of photons for both lateral and axial positioning, specifically 100 photons, resulting in a consistent repositioning time of around 2 ms. Under this conditions, as expected, the precision is not isotropic ($\sigma_x = 34.60 \pm 0.04$ nm, $\sigma_y = 30.36 \pm 0.04$ nm, and $\sigma_z = 39.13 \pm 0.07$ nm). We reported the new results in Figure 2g,h and we modify the Section “Real-time 3D and 4D tracking”.

However, it's important to highlight that our RT-SPT implementation offers flexibility in tracking measurements, allowing for the choice of different dwell times in the axial and lateral directions, either by imposing a priori constraints or by selecting a different number of photons to trigger the re-centering. For instance, a higher number of detected photons along a specific direction can compensate for the lower precision of axial localization. It's crucial to note that while the lateral and axial directions can achieve the same precision using different dwell times, they cannot attain identical spatiotemporal resolution. We highlight this aspect also in the main text:

“An interesting feature of our RT-SPT implementation is the ability to decouple the lateral and axial re-centering by imposing different update rates or different target photon countings for the two dimensions. This allows achieving an isotropic localization precision at the expense of anisotropic re-centering times.”

Upon comparing the precision results obtained from the new induced RT-SPT experiment with those achieved for characterizing the optimal localization volume, we identified an inconsistency. Subsequently, we reanalyzed the static localization precision and identified an error in the source code. The correction has been made, and the revised results are reported in Figure 2a, b, c.

3. When calculating the CRB, the authors make the assumption that background is independent of the excitation intensity based on detector dark noise alone. This is a very poor model for real systems, particularly in cells where the autofluorescence will scale with the excitation intensity. The authors should consider adding in a real background term to the CRB calculation.

We acknowledge the Reviewer's valid point that assuming the background is solely composed of the dark noise from the detector array might not accurately represent various practical experimental conditions. Reflection, ambient light, and out-of-focus fluorescence are significant sources of background, and while the first two can be mitigated with high-quality optical filters and optical enclosures, reducing out-of-focus fluorescence poses greater challenges. Although lowering particle concentration can help reduce this type of background, implementing such a strategy is not always straightforward, particularly when tracking endogenous organelles, e.g., Lysosomes, within living cells.

Given these considerations, it is crucial to incorporate this realistic background source into our theoretical framework to accurately assess the properties of our RT-SPT approach during experiments. Additionally, it's noteworthy that while out-of-focus background increases with the power of the excitation beam, it is independent of the particle's position (like dark noise background). Therefore, it can be appropriately included in the general signal-to-background ratio (SBR) term that we have clearly introduced and described in our theoretical framework (Supplementary Information Note 1). While we could have explicitly introduced a second term for this type of background, we aim to maintain simplicity in notation, aligning with the standards set by prior influential publications on this topic (Balzarotti et al., *Science*, 10:355(6325):606-612, 2017, Masullo et al., *Light Sci Appl*, 11:199, 2022). However, we acknowledge that this aspect could be emphasized more effectively in the manuscript, encompassing both the Main Text and Supplementary Information. To address this, we explicitly state that the Signal-to-Background Ratio (SBR) term incorporates any background source independent of the particle's position, including out-of-focus background (Supplementary Equations S.7 and S.8). Additionally, we draw attention to the fact that the out-of-focus background increases with excitation beam power, underscoring that the SBR is contingent on these specific experimental conditions.

In general, we deeply revised the explanation about the derivation of the CRB to better highlight the assumptions of our model, and therefore improve the clarity of the manuscript. We also changed the following sentence in the Main Text:

“Our calculation relies on three crucial assumptions: (1) both the signal and background photon counts follow a Poisson distribution, (2) the signal exhibits linear dependence on the excitation light intensity, and (3) the background is independent of the particle’s position. The first assumption is applicable to SPAD detectors, the second necessitates low excitation to avoid fluorescence saturation, and the third condition is typically met when the background includes a combination of detector dark counts and unwanted signals from the sample, such as scattering, autofluorescence, and out-of-focus fluorescence.”

4. On page 13, the authors say that: “The discrepancies between the trajectory and the microtubules structure is mostly attributed to the different dimensional of the two datasets, i.e., the trajectory spans across a 3D space, while the image is a single 2D optical section.” Couldn’t it be that the microtubule structure has evolved over time, or that the cell has moved? How long before the trajectory was the “reference image” collected?

We acknowledge the Reviewer's concern regarding potential misalignment between the particle trajectory and the underlying cellular micro-environment, such as the microtubule network. Several factors, including the temporal evolution of the sample, can contribute to this misalignment. Our experimental sequence typically involves the following steps:

- Identifying a region of interest through real-time imaging/scouting of the sample.
- Registering a high-quality reference image of the identified region.
- Performing multiple particle (Lysosomes, in this case) tracking experiments in the registered region.

This sequence may take a couple of minutes, and if the cellular micro-environment undergoes changes during this time, a misalignment between trajectory and structure may occur. In the case of microtubules, their dynamics within the minutes range are relatively limited, and we anticipate that any discrepancy is not likely to significantly alter the qualitative overlapping. However, we emphasize that for quantitative analyses, this discrepancy should be carefully considered. This aspect has been explicitly addressed in the Main Text:

“As expected, the lysosome demonstrates movement along quasi-rectilinear paths, partially aligning with the microtubule structure revealed through imaging. Discrepancies between the trajectory and microtubule structure predominantly stem from the different dimensionality of the two datasets. Specifically, the trajectory spans a 3D space (Fig. 4b and Supplementary Fig. S8), while the imaging captures a single 2D optical section. Additionally, the elapsed time between the recording of the underlying image and the trajectory may contribute to these disparities.”

Reviewer #3 (Remarks to the Author):

Bucci et al. report a new instrument and methodology, based on novel SPAD array technology, for single-particle tracking with an interesting spatio-temporal resolution combination, and the addition of simultaneous measurement of fluorescence lifetime. The paper is clearly written, the results well presented, and the main conclusions are solidly supported by the data. In my view this paper would be ready for publication after the authors address some minor issues.

We greatly appreciate the positive evaluation provided by the Reviewer, particularly with regard to the clarity of our work and, more significantly, the robustness of the results presented. We are pleased that the Reviewer recognizes the novelty of our implementation, which harnesses SPAD array detector technology, showcasing its potential in terms of spatiotemporal resolution and its seamless integration with fluorescence lifetime measurements—a particularly noteworthy aspect. The constructive feedback is invaluable, and we are fully committed to addressing the minor issues raised to ensure the manuscript is ready for publication.

Comments and Questions

1) “the effective achievable tracking range can easily exceed hundreds of micrometres, depending on the dynamical range of the lateral and axial positioners and the field number of the objective lens”. In principle, the sample could also be mounted on a motorized stage, to further extend the tracking range, right? Even if this is not implemented in these experiments, it may be worth mentioning.

The Reviewer is correct in suggesting the use of a motorized stage to move the sample would enlarge the maximum tracking range.

As discussed in the new Supplementary Note 3, the rise time of the positioners is crucial for achieving a high maximum measurable diffusion coefficient. Coherently, we motivated our choice of galvanometric mirrors for the re-centering with their faster response time compared to a piezo-driven devices (see Supplementary Note Figure SN7). The performance of motorized stages is generally even worse than piezo-drive devices and moving the sample concurrently poses the risk actively modifying the behavior of the tracked particle by introducing turbulences and flows in the diffusion medium.

It is worth mentioning that the use of galvo mirrors will cause a non-linearity at large deflection angles. Instead, a motorized stage might not introduce the mentioned bias and would allow an even larger operation range. An hybrid combination of galvanometric mirrors for medium range tracking and servo motors for extending the tracking range might be a thrilling solution, but at the cost of higher complexity.

To clarify this aspect we have added more details in the Supplementary and modified the main text:

“As a result, the effective tracking range of the real-time single-particle tracking is primarily constrained by the lateral and axial scanning capabilities of the microscope”

“Regarding the spatial range, while the OLV is confined to a few hundred nanometers with a high numerical aperture objective lens, the effective tracking range can extend to hundreds of micrometers laterally and tens of micrometers axially. These values are theoretically dependent on the dynamical range of the lateral and axial positioners, the field number of the objective lens, and its working distance. However, practical constraints may also arise due to optical aberrations.”

2) “These assumptions are typically valid in low-excitation experiments, where fluorescence is far from saturation and the primary background source is the detector’s dark noise.” Is this the case in the experiments? Are the dark counts of the SPAD array the main contribution to background? Also, is the assumption about background independent from excitation intensity really necessary? I have the impression that it should not add much complexity to consider a background linear with excitation intensity.

We thank the Reviewer for raising this concern. This assumption on the background is an unusual limitation, and in fact, the same point has also been raised by the second Reviewer (minor issue #2). Therefore, we are copying and pasting our reply here.

“We acknowledge the Reviewer's valid point that assuming the background is solely composed of the dark noise from the detector array might not accurately represent various practical experimental conditions. Reflection, ambient light, and out-of-focus fluorescence are significant sources of background, and while the first two can be mitigated with high-quality optical filters and optical enclosures, reducing out-of-focus fluorescence poses greater challenges. Although lowering particle concentration can help reduce this type of background, implementing such a strategy is not always straightforward, particularly when tracking endogenous organelles, e.g., Lysosomes, within living cells.

Given these considerations, it is crucial to incorporate this realistic background source into our theoretical framework to accurately assess the properties of our RT-SPT approach during experiments. Additionally, it's noteworthy that while out-of-focus background increases with the power of the excitation beam, it is independent of the particle's position (like dark noise background). Therefore, it can be appropriately included in the general signal-to-background ratio (SBR) term that we have clearly introduced and described in our theoretical framework (Supplementary Information Note 1). While we could have explicitly introduced a second term for this type of background, we aim to maintain simplicity in notation, aligning with the standards set by prior influential publications on this topic (Balzarotti et al., *Science*, 10:355(6325):606-612, 2017, Masullo et al., *Light Sci Appl*, 11:199, 2022). However, we acknowledge that this aspect could be emphasized more effectively in the manuscript, encompassing both the Main Text and Supplementary Information. To address this, we explicitly state that the Signal-to-Background Ratio (SBR) term incorporates any background source independent of the particle’s position, including out-of-focus background (Supplementary Equations S.7 and S.8). Additionally, we draw attention to the fact that the out-of-focus background increases with excitation beam power, underscoring that the SBR is contingent on these specific experimental conditions.

In general, we deeply revised the explanation about the derivation of the CRB to better highlight the assumptions of our model, and therefore improve the clarity of the manuscript.”

3) In the first demonstration of 3D tracking, the authors only used the rough estimators (centroid and normalized difference). It would be nice to see how the tracking precision improves when using the MLE estimator. This could be done by off-line analysis of the registered trajectories.

Also for these experiments, it would be nice if the authors mention what the total measurement time was, and discuss the influence of mechanical drift during the measurement.

We appreciate the Reviewer's suggestion, particularly because it prompts a deeper discussion on the spatiotemporal constraints of our tracking method. In the original manuscript, we presented postprocessing offline analysis as a general solution that would consistently enhance the spatiotemporal resolution. However, we recognize that this message was overly optimistic, as the spatiotemporal resolution is fundamentally restricted by particle brightness and dynamics.

In response, we have revised several sections of the main text to emphasize this point. For instance:

"In short, our RT-SPT method leverages the spatial information provided by the SPAD array to continuously re-center the sFoV onto the particle. Like any RT-SPT method, our approach can effectively track a particle only if the system is precise in localizing the particle and fast enough in re-centering the sFoV to prevent the particle from escaping the sensitive region. Indeed, the localization precision scales with the number of photons detected from the particle, whose emission in turn requires a certain period of time depending on the particle's brightness. Because the SPAD array detector has practically sub-microsecond temporal resolution and, for a given microimage, the FPGA can calculate the particle's position in less than 100 ns, the spatiotemporal resolution — i.e., distinguishing two positions of the same particle both in time and space [52] — depends mostly on the lag time of the actuators responsible for sFoV repositioning and the brightness of the particle. Importantly, the SPAD array detector transfers microimages to the PC at a rate significantly higher than the sFoV re-centering rate. If brightness is not the limiting factor, the spatiotemporal resolution can be enhanced with offline analysis. More robust and precise estimators, such as maximum likelihood, can be directly applied to the microimages to achieve a higher localization rate (temporal resolution). However, it is crucial to consider that a higher localization rate comes at the expense of lower photon counts, resulting in lower localization precision. Additionally, the lag time of the actuators may introduce distortions in the distribution of microimages, further impacting the localization precision."

Here we would like to elaborate further on this limitation in order to explain why it is not always possible to increase the spatial or the temporal resolution. To illustrate this point, we refer to the analysis conducted by Rienzo et al. (Spatiotemporal Fluctuation Analysis: A Powerful Tool for the Future Nanoscopy of Molecular Processes. Biophysical Journal vol. 111 679–685 2016). The authors tackle the issue of "dynamic resolution" by examining (1) the localization uncertainty and (2) particle motion as functions of localization time.

Firstly, regarding (1), the precision of our estimation is inversely proportional to the square root of the registered photons (as shown in Equation S.12 in Supplementary Note 1), which, in turn, depends on the localization time and the particle brightness:

$$\sigma_{loc} \sim 1/\sqrt{N} = 1/\sqrt{I \cdot \Delta t_{loc}}$$

As noted by the Reviewer, this parameter has a lower limit independent of photon counting represented by sample and microscope drifts and vibrations (see Rebuttal Figure 10).

Secondly, the displacement of the particle (2) is also clearly time-dependent. Referring to Equation S.19 in Supplementary Note 3, we can express this as:

$$\Delta x \approx \sqrt{\langle \Delta x^2 \rangle} = \sqrt{2n \cdot D \cdot \Delta t_{loc}}$$

Given a specific particle diffusion coefficient and brightness, the minimum localization time (temporal resolution) occurs when the particle's displacement equals the precision of the estimation (spatial resolution).

This observation highlights how the spatial and temporal information are inherently linked. Consequently, the main achievement of the postprocessing algorithm is to trade information between the spatial and temporal domains. The increase of the amount of information content by using more sophisticated algorithms is also possible, but is nevertheless constrained by the aforementioned limitations.

Our technique's postprocessing capabilities are extremely useful to overcome real-time limitations, such as the rise time of the positioners. In scenarios where particle diffusion is slow or the particle is particularly bright, the optimal localization time predicted by the previous model may be shorter than the actuators' rise time. Consequently, the trajectory sampling in real-time becomes suboptimal, necessitating an offline upsampling.

This scenario is exemplified in Figure 3e, f, g, where a fluorescent bead diffuses at approximately $0.5 \mu\text{m}^2/\text{s}$ with an average photon flux of 10 MHz. Furthermore, real-time localization uncertainty is primarily constrained by microscope vibrations rather than the estimation process itself (re-centering occurs every 1 ms, corresponding to 10,000 photons per localization, resulting in estimation uncertainty well below the vibration noise level). Rebinning the trajectory at a sampling time of $20 \mu\text{s}$ yields approximately 200 photons per localization, producing a localization precision just above the vibration noise level. The net effect is a 50-fold improvement on the temporal resolution side, with a negligible effect on the spatial resolution.

Rebuttal Figure 10 - Microscope vibrational noise during tracking. Average drift as a function of time. The curves are obtained by 5 independent experiments tracking 100nm fluorescent beads ($\lambda_{exc} = 488 \text{ nm}$) in an immobile sample for more than 3 minutes. Each acquisition is performed with a fixed re-centering time every 50 ms at an average photon flux of $494 \pm 57 \text{ kHz}$. Adapted from Supplementary Figure S1.

In the case of the Lissajous calibration pattern, the situation is different: we set the number of photons per localization to 100 to ensure comparability with static uncertainty characterization. Given the average photon flux of 50 kHz (see Supplementary Figure S3), this produces a re-centering every 2 ms. With the localization uncertainty associated with 100 photons comparable to the vibration noise level (compare Figures 2e,f with the Rebuttal Figure 10), and the re-centering time already at the real-time limit, the trajectory is acquired under optimal conditions, contrary to the previous scenario.

The Reviewer is nevertheless correct in assuming that some extra spatial information could in principle be extracted without changing the temporal resolution by using, for example, the maximum-likelihood estimator (MLE). In particular, this approach should result in an improvement of about 5 nm (refer to Supplementary Figure S2). However, despite these expectations, attempts to refine the trajectory with the MLE did not yield better precision than real-time measurements. We identify a possible additional source of uncertainty—the "motion blur" effect—capable of overcoming the expected improvement. When a particle moves from point

A to point B within a certain integration window, the resulting microimage deviates from the one expected for a static particle at either point A or point B. The result is a blurred image, hence the term “motion blur”. It's crucial to note that the maximum-likelihood estimator heavily relies on the model it is provided with, in this case, the point spread function (PSF) of the system. Therefore the PSF model is accurate for static localizations, but may fail in the presence of dynamics. In particular, the motion blur leads to miscalibration of the MLE (Rebuttal Figure 11). This affects the postprocessed trajectory. Although this effect could also impact fast estimators during real-time acquisition, it is largely mitigated by the convergence of the feedback loop, resulting in minimal impact on the raw trajectory.

As a final remark, motion blur is not a fundamental limitation in postprocessing. We are confident that a mathematical framework could be developed to account for this effect, drawing inspiration from camera-based algorithms where framerate typically limits time resolution and motion-blurred images pose a significant challenge.

Rebuttal Figure 11 – Motion blur effect on maximum-likelihood estimation. Simulated effect of the motion blur as a function of the particle displacement along the lateral directions (a and b) and the axial direction (c). The simulations are performed using an

experimental PSF ($\lambda_{exc} = 488 \text{ nm}$) to generate ideal microimages which are then integrated to obtain the motion blurred ones. Noise is also added to simulated a scenario with $SBR_p=5$. The same experimental PSF is then used inside the MLE algorithm to retrieve the position from the motion blurred microimages. An additional graph shows standard deviation of the estimation as a measure of the uncertainty. We observe the motion blur introduces a miscalibration while not affecting sensibly the uncertainty.

4) Why don't you show 4D tracking data for the 40 nm beads? Is this a signal level issue? The authors should clarify this and justify the use of the 100 nm beads

In the original version of the manuscript we have selected the 40 nm beads to demonstrate spatial 3D tracking capabilities with relatively low fluxes (similar to biological applications) and then excluded this size when brighter emission was required.

In the new version, however, we have expanded our characterization with also diffusion in free water. The manuscript now displays trajectories of beads of all different sizes (40 nm, 100 nm, 200 nm). For the experiment of the offline analysis (Figure 3e,f,g) we specifically used the bigger bead (200 nm) for the high photostability and relatively low diffusion coefficient.

5) When the authors track the 100 nm beads, they observe a sub-diffusive behaviour that they ascribe to "spatial heterogeneity in the solution". In my view this is rather unlikely. On the other hand, the authors should exclude the influence of optical forces on the larger beads, which may not be high enough to trap the bead but enough to induce a constrained diffusion.

We agree with the Reviewer that subdiffusive behavior is generally unlikely in solutions, and therefore, liquid-liquid heterogeneity might not be the most likely explanation. We appreciate the suggestion of light trapping. Unfortunately, we couldn't rule out this effect as we lack proper characterization of the phenomena for our acquisition. As a precautionary measure, we performed the experiment again and lowered the excitation intensity (new Figure 2e,f). To demonstrate tracking in pure water, we also avoided using glycerol. The lower viscosity and the reduced excitation both suggested using a larger (therefore brighter) bead of 200 nm. The new acquisition does not show any subdiffusive behavior (Supplementary Figure S7), suggesting the effect was caused either by the solution or the excitation level (most likely, since other water-glycerol experiments don't show subdiffusive behavior).

Notably, we concurrently optimized the acquisition towards the measurement of longer trajectories, resulting in tracking the bead freely diffusing for more than 2 minutes.

6) "This behavior is confirmed to a broader extent in the Supplementary Note, where we collect a pool of 15 independent 4D lysosome tracking experiments and analyze them with an automated segmentation algorithm based only on the fluorescence lifetime, hence not requiring any user intervention." The "3" is missing after Supplementary Note. More importantly, it is not clear if the authors analyzed trajectories individually or segments of trajectories. Just before this passage, the authors showed a clear correlation between lifetime and speed (or "run" and "stop" segments). Here, it seems the authors have full trajectories of "run" or "stop" behavior. Please clarify.

We agree with the Reviewer that this experiment is missing clarity in some explanations. Therefore we modified the main text:

"In the proposed case study of single lysosomes tracking, the concurrent measurement of fluorescence lifetime greatly simplifies and improves the recognition of the lysosome motion state. This possibility is further confirmed in the Supplementary SI Note 5, where we segment the motion state of 15 independent 4D

lysosome trajectories by solely analyzing the fluorescence lifetime value in an automatized manner, thereby eliminating the need for any user intervention."

Key changes have been also implemented in Supplementary Note 5, detailing the methodology for lifetime-based segmentation. In summary, all trajectories are initially analyzed altogether to generate an overall histogram of all the measured lifetime values. At this point both space and time are completely ignored. Assuming a correlation between lifetime values and motion states, we establish a probabilistic model to derive an appropriate threshold value. Subsequently, this threshold value is employed to cut the 15 trajectories into "run" or "stop" segments. To validate the segmentation's accuracy, we compute the average mean squared displacement for both categories, consistently revealing faster diffusion in the presumed "run" segments.

Minor corrections

- "Because the information collected by the SPAD array detector is also transferred to the personal computer (PC) in the form of microimages with a high frame-rate, the spatiotemporal precision of the tracking can be improved off-line using a more precise and robust localisation algorithm than the centroid". This phrase makes one think that the recentering is only performed in (x,y). The normalized difference algorithm for the rough axial position estimation is only introduced much later.

We realize it is misleading to refer to the centroid estimator in this part of the manuscript. We revised the text to avoid confusion.

- There are some typographical errors to be considered (e.g. "limit" instead of "limits" in page 3, "different dimensional" in page 13)

We thank the Reviewer for pointing out the typos. We corrected the reported ones and checked the overall text

- Figure 1a there is an error in the legend "Tube lens" and "Cylindrical lens"

We corrected the labeling errors in Figure 1, which now should be readable.

- There are numerous hyperlinks in the text that, at least in my reader, lead to nowhere. Also, in page 10, there is a hyperlink instead of a reference, I think: "we calculate the MSD averaging over all the different trajectories and time intervals as described in 4."

We agree with the Reviewer there is an undesired behavior of the hyperref package in latex. Specifically, it effectively created useless or broken links. We changed the settings and now the text should contain only useful hyper references.

- "The processing pipeline is described in details in the materials and methods supplementary section 4". There is no supplementary section 4

We checked the text and this hyperlink should be solved with the new latex setting (see reply above).

- Fig 3d. The authors should improve the caption and/or main text to explain what are the orange and blue curves

The legend and/or caption description was indeed missing. We added it in the new version of Figure 3g.

- What is the meaning of "The peak value is suddenly recovered on average every 3.4 ± 0.5 s". Does the stop and go have a frequency? It seems to be a rather stochastic process.

We agree with the Reviewer the process looks stochastic and we couldn't identify any remarkable pattern. That number was reported to quantify the average stop time, but we agree it is not particularly informative, especially with that large error bar.

REVIEWER COMMENTS

Reviewer #1 (Remarks to the Author):

The revised version of the manuscript satisfactorily addressed my concerns and questions. Several points have been clarified and substantially improved.

I recommend the manuscript for publication in the present form.

Reviewer #3 (Remarks to the Author):

Bucci et al. present a revised version of the manuscript addressing all comments made by all reviewers. In my opinion, the paper is now much improved and is ready for publication after the authors address some minor comments pointed below.

Minor comments:

1) There are still typos present. For example in the abstract: offline -> offline; an hybrid -> a hybrid; We upgrade – We upgraded.

2) Why is the lifetime of the 40 nm beads so different. Please provide a brief discussion. Also, since the authors noticed that “the fluorescence lifetime seems to exhibit a space-dependent behavior with a region near the beginning of the trajectory associated to a value of around 2.5 ns, which then fades to 3.4 ns as the particle diffuses. However, in this specific example, a deeper understanding is obtained by observing the time evolution of each component of the 4D trajectory”, what is the lifetime reported for each bead in Fig 3b? the initial, the steady-state? Or track average value?

3) In the discussion of Fig 3g, the authors state “The postprocessing pipeline not only adds a 50-fold improvement in time sampling but also reveals previously hidden spatial fast movements that were averaged out in real-time due to the longer integration window”. However, I find it difficult to identify those fast movements. At the scale shown, it seems that the down-sampling only increases noise. Maybe the figure panel of 3g needs to be

improved to show enlarged regions where the increased temporal resolution reveals meaningful faster movements.

4) In the discussion of Fig 4, the authors state “while the imaging captures a single 2D optical section.” The mismatch between the 2D image and the tracking could be larger because the 2D image is actually a projection of the microtubule network corresponding to a diffraction-limited section, whereas the tracking is performed in 3D with sub-diffraction resolution. Suggestion:

“Specifically, the recorded trajectory is in 3D space with sub-diffraction resolution(Fig. 4b and Supplementary Fig. S9), while the 2D imaging captures the projection of a single, diffraction-limited optical section.”

5) In Fig 4c, the sub-diffusive behavior of the lysosomes in the sample treated with nocodazole is not clearly visible due to the vertical scale of the graph. It seems like a linear MSD with a lower slope. Consider plotting this MSD on a different scale.

6) In the discussion of the lysosome tracking data, considering that the lifetime reduction is concomitant with a drop in intensity, the picture that emerges is that the GFP is quenched by some species when the lysosome stops. I think this is a simple explanation of what is happening, more sensible than the 3D rolling proposed in the Discussion section. Of course, identifying the quenching species and mechanism is beyond the scope of this work.

On the topic of the authors' responses to Reviewer #2 in the previous round of reports:

In my view, Reviewer #2 has raised relevant technical points and the authors have addressed them adequately for this publication. The aim of this work is to present a novel single-particle tracking method based on the recently introduced SPAD-array detectors. I think this objective is well met. The capabilities and drawbacks of the SPAD-array tracking are well presented, and the paper is worth publishing. There are always more details to be worked out with a new method, and in many cases the relevance of those technicalities depend on the particular application. Therefore, I am in favor of publishing this work and continue the technical discussion in the community as the method is applied.

Other concerns by Reviewer #2 relate to the interpretation of the lysosome tracking. While those concerns are valid, it is not the point of the paper to delve deep into that problem. Rather, it is to show the capacity of the new tracking method to simultaneously provide position and lifetime information. This capability is well characterized.

REVIEWER COMMENTS

Reviewer #1 (Remarks to the Author):

The revised version of the manuscript satisfactorily addressed my concerns and questions. Several points have been clarified and substantially improved.

I recommend the manuscript for publication in the present form.

I would like to thank Reviewer #1 for appreciating the quality, perspectives, and novelty of this work from the beginning. I also appreciate the extra effort in commenting on Reviewer #2's feedback. I fully share Reviewer #1's opinion regarding the quality of the requests from Reviewer #2, which we were happy and keen to address. At the same time, I agree that this paper is intended to be seminal, and many improvements will come later, as research in life sciences continues to evolve. We appreciate Reviewer #1's insightful feedback on our manuscript and their support for the publication of our novel approach.

Reviewer #1 (Remarks to the Review #2):

Reviewer #2 gave a very detailed review of the manuscript. The reviewer deeply analyzed the text and every detail of the described experiments and the descriptions and interpretations of their respective outcomes. This is clearly an admirable and valuable effort that should be well appreciated. In the end, the manuscript clearly took benefits from the discussion of the points that needed clarification from the reviewers perspective. However, to my opinion a manuscript for a journal that provides only limited space for the authors to convey their research and finding to the audiences one is always forced to make a tradeoff between presenting the new principles and ideas of the research and the level of detail in order to keep the readability and length of the manuscript within a tolerable measure. This leads to the necessity to add extensive amounts of supporting data to provide evidence for the claims given in the article itself.

For me as a researcher it is most interesting to learn new concepts and new experiments and to learn how they were executed and how the ideas of the authors were transferred into an experiment. Technical peculiarities, absolute numbers, or limitations determined by available hardware or theoretical models used for describing the data are of minor interest for me, because one can adapt for these in case needed.

In this sense the comments and remarks of reviewer #2 are all well and good, but they exceedingly stretch the scope of the manuscript (or its supporting information) towards the technical details and are not so much relevant for conveying the basic claims of the manuscript.

To my opinion, the authors addressed the remarks of reviewer #2 in sufficient detail and clarified the manuscript in respect of the points raised by the reviewer.

However, the revised manuscript brought my attention to a discrepancy in the presented data. The discussion of the third of the reviewers remarks led to a set of new experiments in which the authors tracked fluorescent beads with radii of 20 nm, 50 nm, and 100 nm shown in the new figure 3 of the manuscript. Panel c) of this figure shows the MSD with the fitted diffusion constant of these beads with lie all in the expected range for the given sizes. However, the distributions shown in panel a) of the figure clearly show diffusion constants that do not match these numbers.

Is there a real difference between these values, or is this a miscalculation of some sort?

The experiment involving the acquisition of three statistically significant populations of freely diffusing beads was specifically designed to address the inquiries raised by the second Reviewer on various levels. Our aim was not only to characterize the maximum measurable diffusion coefficient, which now experimentally aligns with the estimated theoretical value, but also to evaluate other performance metrics such as tracking length and required intensity. To achieve this, in Figure 3 c and d we selected three trajectories that represented a balance between these parameters, including a diffusion coefficient as expected, a trajectory length as long as possible, and a relatively low photon flux.

While the calculation of the mean square displacement (MSD) is correct, it's important to note that the diffusion coefficient of these trajectories does not necessarily represent the mean of the entire population. In both the main text and the captions, we explicitly state that the trajectories are "three exemplar 4D trajectories, one per bead size, characterized by a tracking time length exceeding 0.5 s and a relatively low photon flux."

However, we acknowledge that one would expect the average diffusion coefficient of each bead population to be compatible with the theoretical value predicted for Brownian diffusion. Nonetheless, our experimental conditions deviate from ideal scenarios due to factors such as bead aggregation and electrostatic interactions with the cover glass, which can reduce diffusion coefficients. Furthermore, as mentioned in the main text, our acquisition parameters were optimized to ensure that tracked particles are lost on average after a couple of seconds, allowing the entire dataset to be acquired within tens of minutes. This condition poses a challenge for faster beads, which are more easily lost and may therefore be underrepresented in the final population.

While we acknowledge that the difference in reported values may initially appear as a discrepancy, we believe that a detailed discussion of this aspect may not be suitable for the main text. The primary aim of Figure 3 a, b, c, d is to demonstrate the feasibility of performing 4D tracking of freely diffusing particles in water up to $10 \text{ } \mu\text{m}^2/\text{s}$. For a more optimized tracking performance, it becomes necessary to focus on acquiring a single trajectory, as demonstrated in Figure 3 e and f.

We hope this explanation clarifies the rationale behind our experimental design and analysis.

Reviewer #3 (Remarks to the Author):

Bucci et al. present a revised version of the manuscript addressing all comments made by all reviewers. In my opinion, the paper is now much improved and is ready for publication after the authors address some minor comments pointed below.

I would like to thank Reviewer #3 for appreciating the quality of this work from the beginning. I also appreciate the extra effort in commenting on Reviewer #2's feedback. I am pleased that Reviewer #3 fully understood the scope and novelty of this paper, as well as its intention to be seminal. We appreciate Reviewer #3's insightful feedback on our manuscript and their support for the publication of our new 4D real-time single-particle tracking method.

Minor comments:

1) There are still typos present. For example in the abstract: offline -> offline; an hybrid -> a hybrid; We upgrade – We upgraded.

We appreciate the Reviewer for identifying the typos. We have addressed the ones mentioned and conducted a thorough review of the text to ensure accuracy.

2) Why is the lifetime of the 40 nm beads so different. Please provide a brief discussion. Also, since the authors noticed that “the fluorescence lifetime seems to exhibit a space-dependent behavior with a region near the beginning of the trajectory associated to a value of around 2.5 ns, which then fades to 3.4 ns as the particle diffuses. However, in this specific example, a deeper understanding is obtained by observing the time evolution of each component of the 4D trajectory”, what is the lifetime reported for each bead in Fig 3b? the initial, the steady-state? Or track average value?

The variation in the fluorescence lifetime of the 40 nm beads can be attributed to several factors. Although they are sourced from the same producer and intended for the same wavelength, differences in their manufacturing process or composition could lead to variations in the fluorophore concentration within each bead. These differences may result in self-quenching or other neighborhood effects, influencing the observed fluorescence lifetime.

Regarding the lifetime reported for each bead in Fig 3b, it is important to clarify that each fluorescence lifetime entry represents the trajectory average. We have included this clarification in the caption to provide better understanding for readers.

3) In the discussion of Fig 3g, the authors state “The postprocessing pipeline not only adds a 50-fold improvement in time sampling but also reveals previously hidden spatial fast movements that were averaged out in real-time due to the longer integration window”. However, I find it difficult to identify those fast movements. At the scale shown, it seems that the down-sampling only increases noise. Maybe the figure panel of 3g needs to be improved to show enlarged regions where the increased temporal resolution reveals meaningful faster movements.

We agree with the Reviewer that the proposed trajectory detail might not have been the best example to support the claim in the text. Therefore we selected another window at a different time which we believe clarifies the point better and with a proper enlargement.

We appreciate the Reviewer's feedback on the discussion of Fig 3g. We acknowledge that the originally chosen figure may not have effectively illustrated the improved temporal resolution as intended. To address this concern, we have selected a different time window from the same trajectory, which we believe provides a clearer demonstration. We trust that this revision will better support the statement in the text and improve the clarity of the discussion surrounding Fig 3g.

4) In the discussion of Fig 4, the authors state “while the imaging captures a single 2D optical section.” The mismatch between the 2D image and the tracking could be larger because the 2D image is actually a projection of the microtubule

network corresponding to a diffraction-limited section, whereas the tracking is performed in 3D with sub-diffraction resolution. Suggestion:

“Specifically, the recorded trajectory is in 3D space with sub-diffraction resolution(Fig. 4b and Supplementary Fig. S9), while the 2D imaging captures the projection of a single, diffraction-limited optical section.”

We thank the Reviewer for suggesting a rephrasing which improves the clarity of the sentence. We changed the main text accordingly.

5) In Fig 4c, the sub-diffusive behavior of the lysosomes in the sample treated with nocodazole is not clearly visible due to the vertical scale of the graph. It seems like a linear MSD with a lower slope. Consider plotting this MSD on a different scale.

We agree with the Reviewer that the original scale may have obscured the sub-diffusive behavior. To address this concern, we have added an inset plot specifically for the nocodazole trajectory. This inset plot utilizes a different scale that better highlights the mentioned sub-diffusion behavior, allowing for clearer visualization and interpretation.

6) In the discussion of the lysosome tracking data, considering that the lifetime reduction is concomitant with a drop in intensity, the picture that emerges is that the GFP is quenched by some species when the lysosome stops. I think this is a simple explanation of what is happening, more sensible than the 3D rolling proposed in the Discussion section. Of course, identifying the quenching species and mechanism is beyond the scope of this work.

We appreciate the Reviewer's observation about possible quenching mechanism occurring during the “stop” phase. We have incorporated this alternative explanation into the discussion section as it appears to be simpler than 3D rolling.

Reviewer #3 (Remarks to the Review #2):

On the topic of the authors' responses to Reviewer #2 in the previous round of reports:

In my view, Reviewer #2 has raised relevant technical points and the authors have addressed them adequately for this publication. The aim of this work is to present a novel single-particle tracking method based on the recently introduced SPAD-array detectors. I think this objective is well met. The capabilities and drawbacks of the SPAD-array tracking are well presented, and the paper is worth publishing. There are always more details to be worked out with a new method, and in many cases the relevance of those technicalities depend on the particular application. Therefore, I am in favor of publishing this work and continue the technical discussion in the community as the method is applied.

Other concerns by Reviewer #2 relate to the interpretation of the lysosome tracking. While those concerns are valid, it is not the point of the paper to delve deep into that problem. Rather, it is to show the capacity of the new tracking method to simultaneously provide position and lifetime information. This capability is well characterized.